



# Zugspitze ozone 1978 – 2020: The role of stratosphere-troposphere transport

Thomas Trickl[1], Cédric Couret[2], Ludwig Ries[2], and Hannes Vogelmann[1]

[1]Karlsruher Institut für Technologie, Institut für Meteorologie und Klimaforschung, IMK-IFU, Kreuzeckbahnstr. 19, 82467 Garmisch-Partenkirchen, Germany

[2]Umweltbundesamt II 4.5, Plattform Zugspitze, GAW-Globalobservatorium Zugspitze-Hohenpeißenberg, Schneefernerhaus, 82475 Zugspitze, Germany

*Correspondence to:* Dr. Thomas Trickl, thomas@trickl.de, Tel. +49-8821-50283; Dr. Hannes Vogelmann, hannes.vogelmann@kit.edu, Tel. +49-8821-183-258

**Abstract.** The pronounced increase of ozone observed at the Alpine station Zugspitze (2962 m a.s.l.) since the 1970s has been ascribed to an increase of stratospheric air descending to the Alps. In this paper, we present a re-analysis of the data from 1978 to 2011 for both ozone and carbon monoxide, extended until 2020 by the data from the Global Atmosphere Watch site Schneefernerhaus (UFS, 2671 m a.s.l.) just below the Zugspitze summit. The analysis is based on data filtering utilizing the isotope [7]Be (measured between 1970 and 2006) and relative humidity (1970 to 2011, UFS: 2002 to 2020). We estimate both the influence of stratospheric intrusions directly descending to the northern rim of the Alps from the full data filtering and the aged ("indirect") intrusions from partial filtering with the [7]Be data. The evaluated total stratospheric contribution to the annual-average ozone rises roughly from 12 ppb in 1970 to 24 ppb in 2003. It turns out that the increase of stratospheric influence is particularly strong in winter. A lowering in positive trend is seen afterwards, almost parallel to the beginning decrease of solar irradiation. The air masses hitting the Zugspitze summit have become drier in the percentile range up to at least 25 % until 2003, and we see the growing stratospheric contribution as an important factor to this drying. Both an increase of lower-stratospheric ozone and a growing width of the intrusion layer departing downward from just above the tropopause must be taken into consideration. Carbon monoxide in intrusions did not change much during the full measurement period 1990 to 2020, with perhaps a slight increase until 2005 and an almost constant behaviour afterwards. This is remarkable since outside intrusions a decrease by approximately 44 % was found, indicating a substantial improvement of the tropospheric air quality.

*Key words:* Ozone, carbon monoxide, water vapour, [7]Be, beryllium, stratosphere-to-troposphere transport, data filtering, transport modelling, lidar, Zugspitze

## 1 Introduction

The rise of ozone at the summit station Zugspitze (German Alps, 2962 m a.s.l.) between 1978 and 2003 has been emphasized in a number of publications during the past twenty years (e.g., Oltmans et al., 2006; Logan et al., 2012; Oltmans et al., 2012; Parrish et al., 2012; Gaudel et al., 2018; Parrish et al., 2020). Until the early 1980s the rise was very strong and associated mostly with growing air pollution. However, during the period afterwards air pollution in Europe stopped and even decreased in the 1990s (e.g., Jonson et al., 2006; Vautard et al., 2006), possibly due the new political situation that led to an improved air quality in Eastern Europe. Scheel (2003)





attributed the continual ozone rise to dry air descending from the tropopause region to the Zugspitze summit. Rising levels of [7]Be hardened the idea of an increasing fraction of stratospheric air reaching the high-lying station. Subsequently, Scheel analysed the stratospheric contribution from the begin of the precision

measurements in 1978 to 2004 (H. E. Scheel, pp. 66-71 in (ATMOFAST, 2005)). The estimated total stratospheric contribution to the Zugspitze ozone more than doubled from 1978 to 2003 (Fig. 1 of Trickl et al. (2020a): 11.3 ppb to 23.5 ppb). Most importantly, the complementary tropospheric ozone contribution no longer exhibited a positive trend. However, the results do not reveal a decline of this contribution since 1990 as one would expect from that of the ozone precursor emissions, which could suggest that this estimate was rather

conservative.

In Fig. 1 we show the development of the monthly mean values for the two summit stations around Garmisch-Partenkirchen (Germany), Zugspitze and Wank (1780 m), from 1978 to 2010 for which the data are evaluated for both stations (the operations at both stations were discontinued after/in 2012, after the retirement of H. E. Scheel). There are several facts:

(1)   The Wank ozone trend stops to be positive after 1981, whereas the Zugspitze ozone continues to increase. We ascribe this difference to the much less pronounced stratospheric impact at the lower-lying station. The number of stratospheric air intrusions reaching the Wank summit is less than 50% of that reaching the Zugspitze summit. (Elbern et al., 1997).

(2)   After 2003 there is an almost parallel, slight ozone decrease at both stations.

(3)   In the 1990s the amplitude of the Wank seasonal cycle diminishes, in agreement with the decreased precursor emission mentioned above. This decrease is less pronounced in the Zugspitze ozone since the higher summit is less exposed to air from the boundary layer. The pronounced summer ozone maxima start to disappear.

There are several questions: Was the increase in Zugspitze ozone until 2003 caused by a change in atmospheric

dynamics due to the climate change? However, if this were the case: What is the reason for the trend reversal after 2003, and is it real? A negative trend was recently reported even until 2017 also for the Italian mountain station Monte Cimone (2165 m a.s.l.; Cristofanelli et al., 2020). How did the Zugspitze trend develop after 2010?

The 1978 stratospheric contribution of 11.3 ppb almost matches the ozone value of just about 10 ppb estimated

for the late 19[th] century (Volz and Kley, 1987; Marenco et al., 1994), in agreement with the idea that stratosphere-to-troposphere transport (STT) was the dominant source of ozone during that early period. However, this value may be too low. Tarasick et al. (2019), based on a large number of publications, suggested a higher background ozone of 20 ppb to 25 ppb during the first half of the 20[th] century. Fabian and Pruchniewicz (1977) describe measurements at the Zugspitze summit between 1970 and 1975. The monthly-mean ozone

values obtained varied around annual means of roughly 20 ppb, with a particularly pronounced positive excursion to about 45 ppb in summer 1970. 20 ppb could mean that the Zugspitze ozone at that time was not far away from the conditions at the beginning of that century. The uncertainties of these data were specified as 1.5 % under clean-air conditions (Pruchniewicz, 1973). However, Tarasick et al. (2019) doubt the assumed quality of these data.

A comparable positive ozone trend is reported for the Swiss Jungfraujoch station (3580 m a.s.l.) where the ozone measurements started in 1992 (Ordoñez et al., 2007). Also in other regions of the world positive trend of ozone





from STT. Clain et al. (2009) derived a positive ozone trend in the upper troposphere above Réunion Island and concluded a growing stratospheric influence also tentatively ascribed to climate change. Cooper et al. (2020) found positive trends for a number of elevated sites world-wide (1971 to 2018), in particular Mauna Loa, and for

IAGOS (In-service Aircraft for a Global Observing System) in the northern hemisphere for a pressure level of 650 mbar.

Despite an altitude of almost 3 km the *in-situ* measurements at the Zugspitze summit can be influenced by air from the planetary boundary layer (PBL) during daytime (e.g., Carnuth et al., 2000; 2002; Yuan et al., 2019). Lidar measurements of water-vapour that showed deviations of the Zugspitze relative humidity (RH) from the

corresponding lidar RH just during warm periods (Vogelmann and Trickl, 2008). Because of the possibility of orographic lifting the data selection within the TOR (Tropospheric Ozone Research) project (Kley et al., 1997) for background conditions excluded related periods. The measurements during periods of subsidence are expected to be fully free of air from the PBL (see also Yuan et al., 2018).

Deep stratospheric intrusions are characterized by very dry air with RH values in the lower free troposphere

frequently far below 1 %. Even for transport times of more than ten days sometimes very little erosion of the layers in tropospheric air is found (Trickl et al., 2014; 2015; 2016). This surprising observation contradicts traditional ideas (e.g., Shapiro, 1976, 1978, 1980). However, there is some evidence also in earlier work that the mixing of descending stratospheric air tongues with tropospheric air could be rather slow (e.g., Bithell et al., 2000; Pisso et al., 2009).

By contrast, the Zugspitze RH typically minimizes at about 10 %, with rare exceptions down to about 3 %. We hypothesized that this deviation is perhaps caused by problems of the dew-point mirror instrument used. This makes the identification of intrusions somewhat difficult. However, Trickl et al. (2010) found by comparison with trajectory modelling that the RH requirements for identifying the impact of STT (stratosphere-troposphere transport) at the Zugspitze summit station are not ultimately critical. A threshold of the RH minimum of 30 %

turned out to be sufficient.

Another interesting finding was that intrusions originate just above the tropopause (Trickl et al., 2014; 2016). The Zugspitze CO values in these layers do not show a massive decrease in these layers in the lowermost stratosphere confirming the idea of a mixing zone around the tropopause (Trickl et al., 2014). This zone seems to be a layer of convergence, collecting contribution from both below and above, the latter also being concluded

from our observations of stratospheric aerosol (Trickl et al., 2013). It is, therefore, difficult to quantify the true tropospheric ozone fraction in this layer.

The seasonal cycle of STT at the Alpine summit stations is characterized by a winter maximum and a summer minimum (Stohl et al., 2000; Trickl et al., 2010). As a consequence, the well-known ozone spring maximum (Monks, 2000) cannot be explained by STT. The minimum is less pronounced at the highest, the Jungfraujoch

station (3580 m a.s.l.), which indicates more STT events reaching higher altitudes during the warm season. In fact, a recent study shows that the STT summer minimum disappears if one looks at the free troposphere as a whole (Trickl et al., 2020a).

This fact and the differences between Wank and Zugspitze ozone mentioned above confirm that the penetration of stratospheric air into the troposphere decreases towards low altitudes. However, the descent of the dry air

tongues continues towards the Mediterranean basin and, thus, a significant stratospheric influence has been evaluated for Monte Cimone (Cristofanelli et al., 2006; 2015; 2020). Sprenger et al. (2003) and Škerlak et al.



(2014) describe the behaviour of deep, medium and shallow intrusions on a global scale and find clear differences. The occurrences of intrusions maximize along the jet streams and deep intrusions are less frequent than medium or shallow ones. The latter is verified by many years of lidar measurements at Garmisch-Partenkirchen (Trickl et al., 2020a). There is little evidence of a penetration of stratospheric intrusions into the boundary layer. Reiter (1990) determined out a total of 1990 ozone, temperature and wet-bulb temperature profiles onboard the Eibsee-Zugspitze cable car (1005 m to 2955 m a.sl.) between 1980 and 1982 and did not observe any case of subsidence of stratospheric layers to below 1.4 to 1.6 km a.s.l. However, the cable car is operated only during day-time, i.e., in the presence of a boundary layer. In fact, Eisele et al. (1999) reported a case of sufficiently deep early-morning descent of a STT layer that it could be caught by the forming boundary layer (see also Schuepbach and Davis, 1994; Ott et al., 2016; Langford et al., 2021).

This paper resumes the Zugspitze ozone trend studies started by H. E. Scheel. Parts of our paper are based on a preliminary manuscript of Scheel, unfinished due his unexpected death in 2013. We completed the data archive for the Zugspitze summit until 2011 based on files with evaluated data found on Dr. Scheel´s computer. No evaluated version of the 2012 data was found. The revised scientific analysis is based on the methods described by Trickl et a. (2010). In order to obtain some idea about the trend development beyond 2011 the study was extended until 2020 with the data acquired at the Global Atmosphere Watch station at the Schneefernerhaus research station about 300 m below the Zugspitze summit.

We first describe and analyse the observational data used and their temporal development. In Sect. 3, we present the characteristics of the data and how they can be used for the data filtering. The filtering criteria eventually used are specified in Sect. 4. Finally, we present and discuss our results in Sects. 5 and 6.

## 2 Observations

### 2.1 Site description

The Zugspitze site of the former Fraunhofer-Institut für Atmosphärische Umweltforschung (IFU; now: Karlsruher Institut für Technologie, IMK-IFU) is located at the northern rim of the Alps (47.421° N, 10.986° E). A detailed description of the topography was given by Reiter et al. (1986). The observations of trace gases, meteorological parameters as well as $^{7}Be$ were made at the summit of the mountain (2962 m a.s.l.) or temporarily close to the summit at 2932 m (1989 – 1994, or during a shorter period, depending on the parameter of the measurement programme). Not all the quantities have been measured over the entire period since 1970, some measurements were abandoned earlier such as $SO_2$ or $CO_2$. The data are available in the data archive of IMK-IFU as half-hour averages, supplemented by statistical products such as daily, monthly and annual means. For the work presented here we use ozone, $^{7}Be$, RH and carbon monoxide.

Since 2001 atmospheric *in-situ* measurements have taken place at the Global Atmosphere Watch (GAW) observatory at the Schneefernerhaus research station (UFS) on the southern face of Zugspitze, operated by the German Umweltbundesamt (UBA, i.e., Federal Environment Agency; 47.417° N, 10.979° E; air inlet at 2671 m a.s.l.). The calibration of the UBA instrumentation is routinely performed and verified as a part of the GAW quality assurance standards. The instruments are controlled daily and serviced on all regular work days. Yuan et al. (2019), in a study on the Zugspitze $CO_2$ time series, characterized the UFS measurements with respect to those at the Zugspitze summit and at a tunnel site in the cliff behind UFS.


Due to the small altitude difference, in principle, the stratospheric influence at UFS should not differ much from that at the summit. However, one should consider that, during the warm season the boundary-layer formation may prevent intrusions to descend much below 3000 m. This has, indeed, been observed by the lidar measurements (Sect. 3). Thus, even local orographic influence can matter and lead to differences due to different upslope winds advecting air from lower altitudes. However, the trends should behave similarly and the results

for UFS can serve as a tool to extrapolate the results for the summit.

### 2.2 Techniques

#### 2.2.1 Ozone

Since the inception of non-wet chemical, continuous $O_3$ monitoring in 1978, this species was measured using a Bendix 8002 chemiluminescence instrument, compared with a portable Dasibi system until 1999 (Reiter et al.,

1987). After parallel operation with UV absorption instruments, the $O_3$ measurements were based on two or three TE 49 analysers (Thermo Environmental Instruments, USA) operated simultaneously at the station. Several comparisons by means of transfer standards ($O_3$ calibrators TE 49 PS) were made with the World Meteorological Organization (WMO) Global Atmosphere Watch (GAW) reference instrument kept at the WMO/GAW calibration centre operated by EMPA, Switzerland (Klausen et al., 2003). The most recent comparison was

conducted in June 2006 and confirmed that the Zugspitze $O_3$ data are on the GAW scale.

At UFS, ozone has been continuously measured by ultraviolet (UV) absorption at 253.65 nm (Thermo Electron Corporation, model TEI 49i) since since 2002. As ozone standard a TEI 49C-PS instrument was used which was calibrated against the ozone standard of UBA (UBA SRP#29) an annual basis. UBA keeps with this standard the German reference normal which was adjusted via BIPM (Bureau International des Poids et Mesures) in Paris to

the valid NIST ozone reference standard which is relevant for the WMO/GAW measurement programme. The measurements were supported by a second instrument (Horiba APOA-370). The instrumentation is fully adequate for Global Atmosphere Watch monitoring. For weekly and monthly calibration a TE 49PS instrument has been used at the station. GAW system and performance audits at the station for surface ozone took place in 2001, 2006 and 2011.

The ozone data for both sites are stored at 0.5-h intervals with an uncertainty less than ±0.5 ppb from the WMO standard (Hearn et al., 1961, see also Viallon et al., 2015). 1-h averages were made available to the World Data Center and the TOAR data base (Schultz et al., 2017). In the present study we use data at half-hour time resolution.

We also present monthly mean values from Fabian and Pruchniewicz (1977) graphically reconstructed from the

figures in that publication.

#### 2.2.2 Carbon monoxide

Carbon monoxide (CO) measurements at the Zugspitze summit started at the end of 1989 with a gas chromatograph equipped with a mercury reduction detector (RGD-2, Trace Analytical). From 1994 onwards this system was supplemented or temporarily replaced by one or two gas filter correlation analysers (TE 48S). Since

August 2004, a vacuum fluorescence instrument (AL5001, AeroLaser, Germany) has served as the primary instrument, which brought about a significant improvement of short-term CO resolution and exhibited a high



reliability. The working standards employed for the calibration were tied to the carbon monoxide scale maintained by NOAA ESRL/GMD (Boulder, USA) through several comparison experiments and are on the scale of the WMO/GAW network. Two different instruments were always used parallel.

At UFS, two gas filter correlation analysers TE 48C and TE48S were used starting in 2002 and substituted in 2004 by AeroLaser AL5001 and Al5002 instruments. The working standards for measurement of CO were adjusted regularly by a group of 6 NOAA laboratory standards to the actual scale of the WMO/GAW measurement network. For the calibration of the AeroLaser instruments two 1 ppm standard CO mixtures from Deuste-Steininger are used. One of these gas tanks is used to calibrate the instrument as working standard,

whereas the other tank is measured as a target cylinder. GAW system and performance audits at the station for carbon monoxide were also carried out in 2001, 2006 and 2011. However, for a number of years the data are not yet finalized.

### 2.2.3 Beryllium-7

Radioactivity measurements were made by IFU at three stations (Zugspitze, Wank, Garmisch-Partenkirchen)

since the late 1950s in view of nuclear fallout (Sládkovič, 1969, and references therein; Reiter et al., 1971). This led to studies of the descent of stratospheric air into the lower troposphere. Routine measurements at the Zugspitze summit of $^7$Be started in 1969 (Reiter et al., 1971) and are archived from 1 January 1970 until 30 April 2006. For the determination of the $^7$Be activity, aerosols in ambient air were sampled on cellulose nitrate filters (Sartorius, No. 11301) using a Digitel DHA 80 high-volume sampler. The filters were found to retain aerosols

with efficiencies between 93.0 % (diameters 0.05 to 0.1 µm) and 99.3 % (diameters > 0.3 µm) (Reiter et al., 1971). The daily filters were measured in a laboratory of IFU by way of high-resolution gamma spectrometry. The sampling time was 24 h, as necessitated by the signal-to-noise ratio, which sets a certain limit to identifying stratospheric air intrusions. Through an intercomparison experiment involving four high-altitude sites in Europe, the Zugspitze $^7$Be results were found to lie within twice the standard deviation of the mean (Tositti et al., 2004).

In 2003 there were two extended periods during which the $^7$Be sampler was out of operation. Since this year was part of a comparison phase with the new DWD $^7$Be sampler the data gaps could be filled with data from DWD. These values are stored twice a day. The periods are time shifted to 7:30 to 19:00 CET and 19:30 to 7:00. This does not matter since the data filtering is carried out on a half-hour basis.

The IFU and DWD $^7$Be data agree rather well. For example, the averages for 2002 are 4.47 mBq m$^{-3}$ and 4.63

mBq m$^{-3}$, respectively, both averages determined on a half-hour grid. This means that the 12-h DWD data were calibrated to match the 24-h IFU data. It could not be clarified why this is the case.

### 2.2.3 Relative humidity

Relative humidity (RH) measurements at the summit with a dew-point-mirror instrument (Meteolabor, model Thygan VTP6) started in 1997 and were officially archived since 1998. The quoted uncertainty is below 5 % RH

For the years 1970 to 1997 RH values from the Zugspitze weather station of the German Weather Service (DWD) were taken (for more information see Sect. 3.4; opendata.dwd.de/climate_environment/cdc/ observations_germany/climate/hourly/air_temperature/, station 05792, the times for this early period are listed in CET rather than the current standard UTC). The archiving interval (1 h) is for these data is longer than the





typical 0.5 h for most of the data of both IFU and UFS data, but the values are listed in our data archive twice per hour in order to match the resolution of our analysis. Also some major data gaps in 2010 and 2011 were filled with RH values from DWD, which turned out to be highly necessary near the temporal boundary (end) of our analysis.

Relative humidity at UFS has been monitored by the German Weather Service with an HMP45D sensor from 3 August 2011 to 15 July 2014 an EE33 humidity sensor (E+E Elektronik) afterwards. There is no information on the sensor type used from the beginning (23 August 2001) to 3 August 2011. We use the data from 2002 to 2020 provided at intervals of 0.5 h, after a conversion of the times from UTC (DWD) to CET.

### 2.2.4 Lidar measurements

Lidar measurements have a great potential for studying stratospheric layers in the troposphere because they are characterized by elevated ozone and very low humidity. Measurements with the IFU ozone (Trickl et al., 2020b) and the UFS water-vapour (Vogelmann and Trickl, 2008) differential-absorption lidar systems have accompanied the *in-situ* measurements over many years. These measurements, together with transport modelling, yielded insight into the descent or long-range transport of the dry stratospheric layers towards Garmisch-Partenkirchen (Eisele et al., 1999; Zanis et al., 2003; Trickl et al., 2010, 2020a) as well as information on the minimum humidity as a function of the transport path length (Trickl et al., 2014; 2015; 2016). Some of these studies included the analysis of long-range transport of air pollution (Stohl and Trickl, 1999; Trickl et al., 2003; 2011). Dense lidar time series at intervals of one to two hours were extended to up to four days and were used to interpret the Zugspitze results. Daily lidar measurements with less dense sequence have been made up to more than one week.

### 2.3 Other tools

Trajectories have been used to verify deep stratospheric intrusions. Many case studies have been made with FLEXTRA trajectories and the particle-dispersion model FLEXPART (Stohl and Trickl, 1999; Trickl et al., 2003; 2010; 2011; 2014). Daily four-day forecasts with the LAGRANTO model (Wernli, 1997; Wernli and Davies, 1997) were provided by ETH Zürich until the end of the ozone lidar measurements in early 2019 (e.g., Zanis et al., 2003; Trickl et al., 2010; 2020a). In cases with subsidence periods exceeding four days HYSPLIT (Hybrid Single-Particle Lagrangian Integrated Trajectory, Draxler and Hess, 1998; Stein et al., 2015; https://www.ready.noaa.gov/HYSPLIT_traj.php) 315-h backward trajectories were calculated. We prefer the "reanalysis" mode that have better explained our observations.

Linear regression analyses were made by applying a program developed for spectroscopic studies (e.g., Trickl and Wanner, 1984; Trickl et al., 1993; 1995). This program includes error propagation and recalibration of the standard deviations based on a $\chi^2$ analysis. In addition, more than 20 programs were developed to convert data formats, to fill gaps in the data archive (including import data from external sources), to calculate percentiles of [7]Be and RH, and for the data filtering. For sliding arithmetic averages we use the tool provided by the ORIGIN graphics program used to prepare the figures. This tool generates rather realistic boundary conditions at both ends of the data sets to be smoothed.





**3 Characteristics of the parameters used for data filtering to identify STT**

Reiter et al. (1977) pointed out elevated levels of Zugspitze ozone exceeding the U.S. Federal standard of 80 ppb occur during situations favourable for dry intrusions. A value of 145 ppb was registered during a high-ozone case on 8 and 9 January 1975. Sladkovic et al. (1994) found that both high and low levels of $O_3$ in spring and summer (1984 – 1993) were associated most frequently with northerly winds. The strongly descending air

masses were characterized by elevated specific activities of $^7$Be. From the Zugspitze radioactivity measurements of $^7$Be and $^{32}$P stratospheric residence times of ≥36 d during spring and ≥17 d during autumn were estimated (Reiter et al., 1975).

In the following sections, some characteristics of the parameters employed for data filtering are presented as far as these details are of relevance for the subsequent ozone analyses. In previous efforts the STT influence on the

Zugspitze ozone has been determined by data filtering based on correlating ozone, water vapour and $^7$Be (Elbern et al., 1997; Stohl et al., 2000; Scheel, 2003; Trickl, 2010). The STT fraction strongly depends on the threshold conditions chosen (Stohl et al., 2000). The $^7$Be threshold used at IFU in studies until 2000 is 8 mBq m$^{-3}$ (24 h)$^{-1}$ (Sládkovič and Munzert, 1990). This value was also used by Stohl et al. (2000). Elbern et al. (1997) applied variable thresholds, given by a pre-defined increase of the standard deviation of the values against the running

monthly mean, for the species used. In a study of stratospheric intrusions at Mt. Cimone, Cristofanelli et al. (2006) employed both the fixed value of 8 mBq m$^{-3}$ and a dynamic threshold based on running monthly means. By comparison with trajectory-based predictions of stratospheric air intrusions for the period 2001 – 2005, Trickl et al. (2010) showed that the $^7$Be criterion could be weakened to a threshold of at least 5.5 mBq m$^{-3}$.

The specific activity per intrusion has changed over the years which must be accounted for. As a consequence,

Scheel (2003) replaced the 8 mBq m$^{-3}$ threshold by the 85$^{th}$ percentile, which was also adopted by Trickl et al. (2010) in one of their criteria. Scheel applied a combination of this value with 60 % relative humidity for identifying periods of STT (pp. 71 in (ATMOFAST, 2005); figure 2.45 also published as Fig. 1 by Trickl et al., 2020a). The 60 % RH threshold is a reasonable choice to identify the relevant time period of the humidity trough created by a stratospheric air intrusion (Fig. 2, marked by horizontal arrows). 60 % roughly defines the full width

at half maximum of an intrusion event. The RH increases below this level and the losses above this level, both caused by tropospheric mixing, approximately compensate. Thus, for ozone this interval represents rather well the stratospheric contribution if the centre of the intrusion reaches the summit station. This is sometimes the case and infers uncertainty.

Trickl et al. (2010) introduced an additional RH = 30 % threshold that ensures that sufficiently dry air is detected

at least somewhere within the intrusion layer. Beekmann et al. (1997) took an RH threshold of 20 %.in their analysis of ozone sonde data. The higher value for the Zugspitze filtering was found to be adequate, as verified by comparisons with the predictions by transport models (Trickl et al., 2010).

In Fig. 2, time series of several key constituents measured at the Zugspitze summit are displayed during a period well characterized by ozone lidar measurements and transport modelling (Trickl et al., 2010; 2011). The lidar

time series demonstrated the intrusion fully descended across the summit. The mutual correlations give clear hints on atmospheric long-range transport. On late 21 July 2001 an intrusion from Greenland descended across the summit station until the next morning, exhibiting minimum RH = 7.19 % at 7:00 CET (Central European Time = UTC +1 h), minimum NO$_y$ = 0.07 ppb at 4:00 CET and maximum ozone = 70.76 ppb at 2:00 CET. The





Munich radiosonde (1 CET) showed a much smaller minimum RH of 3 % which is in better agreement with the expectation for the calculated downward travel time of 8 to 9 days (Trickl et al., 2014; for more information see Sect. 3.4). The 24-h $^7$Be value is 8.26 mBq m$^{-3}$, exceeding the classical threshold of 8.0 mBq m$^{-3}$. For CO just a small concentration dip is observed during the intrusion which is rather typical. CO turned out to be not a good tracer of stratospheric air. Trickl et al. (2014, 2016) explain this by the fact that intrusions emerge from the lowermost layer of the stratosphere where obviously the descent to stratospheric values (20 to 40 ppb, e.g., Zahn

et al., 1999; Fischer et al. 2000; Pan et al., 2004; Hegglin et al., 2009; Vogel et al., 2011) is not yet pronounced. Of course, CO has been used to identify polluted air masses. It is interesting to mention that the highest CO mixing ratios have been observed in the volume of fronts (R. Sládković, personal communication). A strong rise of CO in a front is frequently associated with polluted air picked up over industrial region to the north west. In Fig. 2, this frontal passage took place on 20 July 2001, but with just a moderate increase in CO and NO$_y$.

On 23 July a minimum of all species but RH and $^7$Be occurred. This period is characterized by advection from the boundary layer over the tropical Atlantic, a typical behaviour shortly after the beginning of high pressure (Trickl et al., 2003; 2010). NO$_y$ was even lower than in the intrusion.

An important question is: where does the elevated $^7$Be on 24 July and on the following days come from? Apart from three RH dips around midnight between 23 and 24 July the RH values do not indicate any dry layer. If the

elevated $^7$Be observed during this period originated in the stratosphere the air mass had almost completely lost its characteristics. Indeed, HYSPLIT backward trajectories run over 315 h for start times on late 24 July and early 25 July indicate a long-range descent from high altitudes over Arctic Canada, starting before that computational period.

It is obvious that the only identification of stratospheric air from observational data is possible for direct

intrusions into the lower troposphere, i.e., intrusions that descend to altitudes above the boundary layer within approximately 3 to 15 days during which the layers stay dry. For indirect intrusions an estimate can be made based on assumptions on the stratospheric fraction of $^7$Be.

### 3.1 Ozone

A clear rise of ozone during the relevant time period is a good indicator of stratospheric air. The intrusion in Fig.

2 is a good example. However, pronounced ozone peaks are not always the case. There are several factors that can lead to less pronounced signatures. Of course, aged intrusions undergo mixing with tropospheric air, which could be verified by humidity measurements by the UFS water vapour lidar or radiosondes. As mentioned, we found that just the lowermost layer above the tropopause starts to subside (Fig. 18 of Trickl et al. (2014)). This is mostly the case in winter and sometimes leads to just a small rise in ozone (Trickl et al., 2020a). Finally, there

are cases in which just an edge of the intrusion layer hit the station which can be associated with lower peak ozone. Small rises in ozone are hard to distinguish from the mostly rather variable concentrations around an intrusion period. Thus, we exclude ozone from the list of parameters used for identifying STT. This is justified by the thorough analysis by Trickl et al. (2010).

As discussed by Parrish et al. (2020) there is just a small average relative decrease of 2.6 % between the ozone

values at the summit and the lower-lying UFS station. Therefore, we use the UFS data to extend the time series after 2011. The question is how well the results for the stratospheric influence at UFS match those for the





summit during the period of overlap. In any case, even if there is a discrepancy at least the trend for the extended period of observation can be judged.

The values of Fabian and Pruchniewicz (1977) for 1970 to 1977 look somewhat low and are not included in our
analysis. Instead, we use an extrapolated constant ozone mixing ratio for the period 1970 to 1977 as justified in Sect. 5.2.

In general, short gaps of ozone values are filled during the analysis by the respective monthly average.

### 3.2 Carbon monoxide

Carbon monoxide has both natural and anthropogenic sources (e.g., Duncan et al., 2007) and is of importance in
tropospheric chemistry because of its reaction with the OH radical (Logan et al., 1981). The Zugspitze CO mole fractions display a pronounced seasonal cycle with a maximum around April, reflecting the stronger anthropogenic sources in winter, and a minimum from summer to autumn.

For the STT studies they have yielded a guess on the CO values just above the tropopause (Trickl et al., 2014). These values are just slightly lower than the tropospheric average contribution at the Zugspitze summit. Trickl et
al. (2014) determined a very small positive CO trend in intrusions for 1900 to 2005. This trend is not significant and just caused by a locally elevated annual average for 2003 of 133 ppb and a small value in 1990 of 117 ppb. The CO trend outside intrusions during that period is slightly negative indicating an improving air quality.

Here, we repeat the analysis based on one of the revised filtering criteria and extend it to 2020 by including the UFS data that start in 2009.

**3.3 Beryllium 7**

As strongly elevated values of the $^7$Be specific activity [mBq m$^{-3}$] are indicators of stratospheric intrusions (Reiter at al., 1983), this tracer has been important for flagging ozone data with respect to stratospheric influence. Its half-life time of 53.42 d ± 0.1 d (Huh and Liu, 2000) is rather suitable for studies of STT. There are two drawbacks that limit the specificity of $^7$Be. As mentioned, one is the small concentration of the isotope in
intrusions necessitating sampling over 24 h in the apparatus used. Secondly, $^7$Be is not only produced in the stratosphere, but also in the upper troposphere with an estimated contribution of about 33 % on global average (Table 3 of Lal and Peters, 1967). This fraction does not apply for our latitude of about 47.5º N, where just 23.4 % is obtained from Fig. 16 of Lal and Peters (1967). It is even smaller at the higher altitudes of typical source regions relevant for the observations of descending stratospheric air at the northern rim of the Alps. For example,
we estimate from the same figure at 70º N and higher latitudes a constant tropospheric fraction of just 10 %.

The primary production mechanism of stratospheric tracers such as $^7$Be, $^{10}$Be and $^{14}$C is spallation or neutron capture by cosmic rays or solar wind (Lal and Peters, 1967; Herbst et al., 2017). The atmospheric production of these isotopes is modulated by the solar magnetic field, solar wind and the geomagnetic field strength. For the period covered here, the influence of nuclear testing can be ruled out since the last atmospheric nuclear test took
place on 29 September 1969 (in China; https://en.wikipedia.org/wiki/Nuclear_weapons_testing), given the short life time of the $^7$Be isotope.

Figure 3 shows the full $^7$Be time series from 1970 to 2006, together with gliding 90-d and 365-d arithmetic averages. An almost steady increase is visible after 1976. Reiter (1973a; 1973b; 1979) and Reiter and Littfaß





(1977) point out the importance of solar flares in the $^7$Be data. However, the averages shown are obviously not strongly correlated with frequency of solar flares (source: https://www.ngdc.noaa.gov/stp/space-weather/solar-data/solar-features/solar-flares/index/flare-index/) and annual sun spot numbers (source: http://www.sidc.be/silso/DATA/SN_y_tot_V2.0.txt) also displayed in Fig. 3. The increase in $^7$Be since 1977 is not clearly correlated with the frequency of solar flares or the number of sun spots per year. It seems that the solar activity decreases in the new millennium, but $^7$Be does not diminish. We conclude that the observed $^7$Be

values are reasonable proxies for STT. However, it is reasonable to use an additional tracer such as RH for identifying intrusions.

Indeed, Herbst et al. (2017; Fig. A1) found that the global production of $^{10}$Be in the atmosphere for 1960 to 2015 predominantly consisted of a constant term plus an eleven-year modulation with an amplitude of just roughly 20 % of the constant level.

Since $^7$Be is attached to aerosols, it is subject to washout, which might mask the original stratospheric signature. However, Reiter et al. al. (1971) found that washout is relatively small for air directly transported downward to 3000 m from the tropopause. Stratospheric air directly transported to the Zugspitze summit is usually accompanied by clear-air conditions nicknamed "$^7$Be weather" (Eisele et al., 1999). However, washout could play some role for the isotopes reaching the station after long-range transport. It is interesting to note that for

low-lying stations the tropospheric life time is estimated as 35 days, including washout (Bleichrodt, 1978).

As we know from our aerosol lidar measurements the role of pick-up of $^7$Be by aerosols in the free troposphere could be limited by the frequently very clear conditions in this altitude range, whereas there is a persistent aerosol layer in the lower stratosphere and the tropopause region that was particularly pronounced between 1980 and 2000 due to major volcanic eruptions (Jäger, 2005; Trickl et al., 2013). However, in Fig. 3 there is no

evident positive correlation of the $^7$Be data with the extreme eruptions of El Chichon (1982) and Mt. Pinatubo (1991). Thus, quite unexpectedly, the amount of stratospheric aerosol does not play a major role. Given the clean air in intrusion layers it is unclear where the $^7$Be atoms get attached to aerosol.

The winter minimum of $^7$Be in 1970 seems to show the local influence of weather. That winter was characterized by never-ending snowfall resulting in 4 m of snow in a neighbouring valley by the beginning of spring. In the

early 1970s $^7$Be was slightly elevated with respect to the later minimum. It will be interesting to see if this is reflected by the intrusion activity.

In Fig. 4 we show four examples of annual distributions of $^7$Be at time intervals of ten years. The daily data are re-organized in time according growing with size as needed for calculating percentiles. In all four years the maximum number of measurement days is less than 365 or 366 days as reflected by the changing position of the

rise to the highest values. the curves are rather smooth. For later years their slopes become steeper than at the beginning of the measurements in the 1970s.

Figure 5 shows the series of annual percentiles for the entire period 1970 to 2006. The highest values, representing also the highest specific activities in single intrusions, change as a function of time. This behaviour is also seen for smaller values. Thus, it is reasonable to use percentiles as thresholds for the stratospheric origin

of an air mass.

Stratospheric air may arrive at the Zugspitze summit with much longer travel times than the 2 to 15 days found for detectable intrusions in our previous analyses. This component can no longer deliver a similarly clear signature in the data, with the exception of $^7$Be. For the longer travel times, in principle, the limited life time of



the isotope must be taken into consideration, but this is a difficult task in absence of information on the respective transport path and time. In addition, we would, in principle, need to know the stratospheric fraction at the source, not at the receptor site. We, therefore, decided to use percentiles also for determining the indirect stratospheric component in the observations.

### 3.4 Relative humidity

The signature of air influenced by stratospheric influx also comprises a pronounced decrease of humidity during the respective episode. Stohl et al. (2000) have discussed the advantages and drawbacks of the parameters specific humidity and relative humidity. In conclusion, RH is preferred, and it may well serve for identifying stratospherically influenced air, at least for a fixed altitude. Indeed, Trickl et al. (2010), based on trajectory forecasts and backward trajectories, found a very high probability of identifying intrusion layers reaching the Zugspitze summit if, additionally, a minimum RH < 30 % was fulfilled in a given layer. However, in principle it is not an unambiguous stratospheric air tracer. A combination of a RH criterion with a [7]Be criterion is the best choice.

Figure 6 shows monthly percentiles of the Zugspitze RH from 1978 to 2011. An obvious drying of the lower free troposphere during that period is indicated. However, also deeper subsidence in more recent years must be taken into consideration.

The monthly minimum RH can be as low as 1 %, mostly during the cold season, but is higher between 1985 and 1997. This suggests the presence of three phases with perhaps different sensors. As pointed out above DWD data are used for the period 1970 to 1997. Indeed, according to DWD listings there was a change in sensors on 13 March 1986 (from a psychrometer to a MIRIAM-TDH system). This suggests that the missing winter minimum in 1985 must be accidental. The step does no longer exist for the 5[th] and higher percentiles.

For the data filtering we correct the data during the period from 13 March 1986 and the end of 1997 by applying the formula

$$RH_c = RH - \Delta_{RH}(\frac{RH-100}{\Delta_{RH}-100})^8 \qquad (1)$$

$\Delta_{RH} = 5$ (all in per cent). This formula significantly modifies the RH values for low RH, by −0.43 %, −1.26 % and −3.24 % RH for 30 %, 20 % and 10 % RH, respectively. It represents just an estimate, but should not lead to a significant enhancement of the uncertainty of the RH measurements of several per cent.

In general, the Zugspitze relative humidity is dominated by high values, mostly 100 %. This amplifies the detection capability for subsiding air masses that are associated with clear weather conditions. The explanation is the frequent formation of orographic updraft and cloud formation at and above this isolated high mountain.

Despite this step it is obvious that there is a pronounced downward trend also at higher percentiles. Figure 7 shows 25-month averages for several percentiles, roughly representing a one-year time resolution (e.g., Trickl et al., 2020b).

In Figs. 8 and 9 we show the corresponding percentile development for UFS. In Fig. 9, the negative trend in RH seen in Fig. 7, after some stable years from 2004 to 2008 continues until 2011. Afterwards, no trend is observed at all.





Between 2001 and late 2011 the RH minima at UFS are about 7 % RH. This suggests the use of a different type of humidity sensor, but we do not have clear information on an instrument change. We apply Eq. 1 until 31 July 2011 (Sect. 2.2.3) with $\Delta_{RH}$ = 4.45 %. This choice leads to similar minima as for the period after 2011. We do not correct the RH values during the entire period to yield minima around 1 % for 2002 to 2020 or less because this would be speculative. In any case, the correction for 2001 to 2011 did not result in a change in the data

filtering result.

In two of our earlier papers (Trickl et al., 2014; 2016) we document that deep intrusions are much drier than measured at the Zugspitze sites, for both the lidar and sonde measurements. We are now astonished to see RH minima in the summit data of 1 to 2 % in many years, even for different sensors. This suggests that the hypothesized instrumental wet bias of the RH sensors under dry conditions does not exist or is much smaller.

These minima mostly occur during the cold season. As a matter of fact, the UFS DIAL revealed in general deviations from the measurements at the summit just during warm and convective conditions (Vogelmann and Trickl, 2008). We conclude that orographic transport takes place in a shallow surface layer that influences the humidity measurements at the summit station and UFS, but not in the lidar measurements that probe the humidity outside this surface layer. Also evaporation of moisture from the slopes and terraces around the stations

could contribute to the elevated humidity minima.

In 1991, lidar measurements revealed that around noon during the warm season the aerosol level at 3 km and higher suddenly rises due to orographic upward transport emerging from an upvalley air flow followed by backstreaming aloft during day-time (Carnuth et al., 2002). The air stream in the upper elevations, returning from the mountains, is characterized by a delay of the arrival of the aerosol with respect to that in the valley.

This effect was confirmed in more detail during a field campaign in the Swiss Mesolcina valley in 1996 (Furger et al., 2000; Carnuth and Trickl, 2000).

Apart from the build-up of the return flow a direct shallow layer directly creeps up to the summits and crests around a valley already in the early morning (Fig. 5 of Carnuth and Trickl, 2000). The Zugspitze summit could act like a chimney focussing directly rising air at least slightly into a dry stratospheric layer which could be the

reason for the positive bias under warm conditions. In principle, the measurements at the summit station during periods of subsidence should be the least affected by air-mass mixing caused by orographic effects, which is indicated by the lower minimum RH values in winter. We examined the typical occurrence of RH values ≤ 5 %. These cases predominantly occur during night-time.

Mixing is much more likely to influence the results at UFS which might explain the higher minimum values.

Yuan et al, (2019) examined the diurnal cycles of $CO_2$ at three Zugspitze sites, UFS, a tunnel above UFS and the summit station and found significant differences of orographic influence. Of course, the probability of an intrusion to overlap fully with the station is lower at UFS than at the summit due to the limited penetration of the dry layers towards the ground.

In summary, the orographic admixture of humid air is low to moderate. Thus, our analysis of STT should be

rather realistic.

### 3.5 Lidar measurements

Lidar measurements at IMK-IFU and UFS have not been made throughout the year and around the clock. However, a large number of intrusion cases have been studied and allow us to draw important conclusions.





A good example for an intrusion case analysed with both the UFS water-vapour DIAL and the Zugspitze summit
*in-situ* data is given in Figs. 10 and 11 of Trickl et al. (2016). The lidar figure shows the descent of the extremely
dry layer exhibiting a stratospheric-type water-vapour minimum mixing ratio (about 0 to 50 ppm or RH << 1 %)
across the Zugspitze summit, and the station time series verifies elevated ozone up to 73.3 ppb. The Zugspitze
RH minimized at 7.2 %, i.e., substantially higher than suggested by the lidar measurements (see Sect. 3.3).

A complete downward passage of an intrusion layer across the summit station was verified with the two DIAL
systems in many other measurement series. However, the lidar measurements also show examples for intrusion
that do not descend to altitudes below 3000 m, mostly during the warm months. Sometimes these layers persist
over several days at rather constant altitude, typically around that of the Zugspitze summit.

Intrusions do not necessarily lead to a pronounced rise in ozone (Trickl et al., 2014), especially during the cold
season (Trickl et al., 2020a). The upper panel of Fig. 10 shows lidar measurements of ozone on 3 to 7 October
1997 at intervals of one hour. The intrusion structures below 4 km are not as clearly visible as in other colour-
coded images presented by us. The small rise in ozone is verified in the lower panel of Fig. 10 by the
corresponding Zugspitze time series that exhibits just four 60-ppb peaks residing on a 50-ppb background. On 4
October the overlap with the intrusion is limited, however resulting in some rise of [7]Be. On the following two
days [7]Be is substantially higher, but the RH minima are just slightly below 30 %. It is important to mention that
the RH minima in the Munich, Hohenpeißenberg and Stuttgart radiosonde data on these days are much lower
and range between 5 and 15 %, sometimes at just slightly higher altitudes. Also on the other side, to the south-
west, the Innsbruck radiosonde verifies much drier conditions. HYSPLIT backward trajectories initiated at the
Zugspitze summit show long descents from high altitudes over more than 10 days, in part from Siberia. Mixing
with surrounding tropospheric air is likely during this long travel of the layer. Please, note that the two most
pronounced RH dips in Fig. 10 occurred around noon when the orographic wind system maximizes. This could
be the reason of the rather high minimum RH.

We conclude from the lidar results that the intrusion air masses fully hitting the Zugspitze summit are much
more stratospheric than suggested by the *in-situ* RH measurements. This was established from many years of
comparing the *in-situ* humidity measurements with lidar and sonde profiles. However, a bias of unknown
magnitude can be introduced by intrusions just partly overlapping with the summit, i.e., with the humidity
minimum located above 3 km. Such a bias cannot be determined from the available data alone. However, these
cases are more likely to occur in summer when less cases are registered.

## 4 Filtering criteria for quantifying STT at the Zugspitze sites

The re-analysis of the STT fraction in Zugspitze ozone is based on the findings of Trickl et al. (2010). The three
filtering criteria of Trickl et al. (2010) are:

*Criterion 1:* The [7]Be value corresponds to more than the 85th percentile with respect to all data in the respective
year and RH < 60 %.

*Criterion 2:* RH < 60 %, and RH < 30 % for at least one of the half-hour averages within ±6 h. The second
threshold is added to guarantee really dry conditions as expected for stratospheric air.

*Criterion 3:* Same as Criterion 1, but with 5.5 mBq m$^{-3}$ as the threshold for [7]Be.



The application of these criteria yield a rather reliable identification of stratospheric air layers, as verified by transport modelling (Trickl et al., 2010). Daily trajectory forecasts of intrusions were used (Zanis et al., 2003) that also include a coarse altitude information. The number of trajectory bundles forecasted to hit the Zugspitze summit were higher than the number of intrusions identified with data filtering. This could be due to the filter criteria chosen, but also to uncertainties in the coarsely presented vertical positions of the trajectories in the case of missing full overlap of an intrusion with the summit. In the case of forecast gaps or descents over more than four days also HYSPLIT backward trajectories based on re-analysis data were used.

There are differences in the identification of intrusions. For example, Criterion 1 yielded just less than one half of the intrusion cases predicted by transport modelling during the period 2001 to 2005. Criterion 1 was that yielding the result of Scheel obtained in 2005 (Trickl et al., 2020a). The application of Criteria 2 and 3 was significantly more successful. However, there were seasonal differences. Criterion 2 was less successful during the warm season, but yielded more cases in winter. In addition to enhanced orographically induced mixing there is sometimes a less complete overlap of intrusion layers with the summit station, verified in many cases by lidar measurement. This is an important issue since the edges of a layer is likely to contain more tropospheric air than the centre of the layer. If the full layer descends over the air inlet of the summit station the stratospheric contributions can be reasonably well approximated by the selection of the 60-% RH threshold. However, descent of an intrusion layer is only obvious during the early period. Later on, even ascent is possible, e.g., by a rising boundary layer. In one case the lidar measurements revealed (May 2008) thermally driven ascent during daytime and decrease in ozone at the Zugspitze level and descent during the following night and the return of elevated ozone.

In this paper we exploit modified Criteria 2 and 3 to refine the preliminary result of Scheel for 1978 to 2004 (Fig. 1 of Trickl et al., 2020a). In the re-analysis we replace the 5.5 mBq m$^{-3}$ threshold by the 65$^{th}$ percentile of the annual data for 2001-2005 (Fig. 5). We replace the 60-%_RH threshold by 50 % since this eliminates many very thin RH dips. Furthermore, we ignore (or interpolate) strange data that exist for very short periods, if their occurrence does not exceed 2 h. Such data are, e.g., RH values near 100 % interrupting an intrusion air flow reaching the summit, which we tentatively ascribe to fog or clouds ascending from wet slopes. Finally, we search for $^7$Be above the threshold within ±12 h, and RH values below the 30 % threshold within ±15 h, respectively. The additional requirement for $^7$Be was introduced in order to identify intrusions periods beyond midnight.

Unfortunately, there are no $^7$Be measurements at UFS. Thus, data filtering for UFS is confined to the RH criterion. We compare the results for both Zugspitze site during the period of simultaneous measurements.

The $^7$Be data allow us to estimate the contribution of the indirect stratospheric ozone component, i.e., ozone that cannot be evaluated by the data filtering. As explained in Sect. 3.2 we derive the contributions for assuming that 80 % or 90 % of the beryllium is produced in the stratosphere. The $^7$Be specific activities $t_{low}$ for these thresholds are determined by integrating the $^7$Be percentile curves for all years as those shown in Fig. 4. This is done on the generally used half-hour time grid to allow for normalization. Just the sum of the values between the 65-% threshold $t_{dir}$ (labelled by "threshold" in Fig. 4) chosen for the direct intrusions. In addition, we add the half-hour values on days with exceedance of $t_{dir}$ if on these days RH > 50 %. In this way we avoid doubly counting intrusions. In summary:

$$\left[O_{3,indir}\right] = \frac{\sum\left[O_{3,dir}\right]}{\sum\left[^7Be_{dir}\right]}\left(\sum_{\left[^7Be\right]\geq t_{dir},\,RH_d\geq 50}\left[^7Be_{indir}\right] + \sum_{\left[^7Be\right]\geq t_{low}}^{\left[^7Be\right]<t_{dir}}\left[^7Be_{indir}\right]\right)\left(n_{tot}\right)^{-1} \qquad (2)$$



The quantities in square brackets are mixing ratios or specific activities. $n_{tot}$ is the number of half-hour bin with valid data in a given month or year, respectively. $RH_d \geq 50$ % means all RH values on a given day must be greater or equal than 50 %, which avoids including half-hours during a day on which already a direct intrusion was identified. The sums are carried out over both a single month and a single year, respectively, fulfilling the specified restrictions. The conversion of the $^7Be$ sums related to indirect events into ozone is achieved by

applying the evaluated monthly or annual averages of the $O_3/^7Be$ ratios for direct intrusions, for the sums over a single month or one year, respectively. This is justified since both species undergo the same mixing process.

## 5 Results of the data filtering

An important question is what has determined the growth of ozone due to stratospheric influence at the Zugspitze summit. It could be an increase of the number of intrusions per year, the growth of the average length

of an intrusion or an increase of the ozone per intrusion, or a combination of all. We examined all three possibilities.

### 5.1 Intrusion count

In a first step we determined the monthly and annual average number of intrusions. In Fig. 11 we present the number of intrusions per month based on the $^7Be$ criterion (January 1970 to April 2006). We exclude short

events of $\leq 2$ h. A moderate rise of the intrusion rate of is seen. A linear least-squares fit of the data yields a rise of 0.0456 $a^{-1}$ the intrusion count (standard deviation 0.028 $a^{-1}$). Figure 12 displays the same for the RH criterion (slope: 0.0474 $a^{-1} \pm 0.020$ $a^{-1}$), together with the regression line for the $^7Be$ criterion from Fig. 11. It is interesting that the slopes of the two regressions are almost equal. The elevated relative standard deviations are caused by the strong variability of the monthly averages that served as the input data of the least-squares fits. For

the $^7Be$ criterion the calculated count grows from 3.57 (1970) to 5.21 (end of 2005), i.e., by 46 %. For the RH criterion the growth is 2.56 (1970) to 4.17 (2006) and 4.44 (2012) (by 63 % and 73 %, respectively).

The red curves in Figs. 11 and 12 represent sliding $\pm 12$-month averages, which means a single-year temporal resolution following the definition in a VDI guideline (VDI, 1999; see Iarlori et al., 2015; Leblanc et al., 2016, and Trickl et al., 2020b for other definitions). The residual noise is explained by the fact that the arithmetic

average is not a perfect frequency filter.

As already shown for 2000 to 2004 by Trickl et al. (2010) there is a summer minimum and a winter maximum in the monthly intrusion count. This is verified now for the entire period of the Zugspitze measurements. In Fig. 13 the monthly counts for all years 1978 to 2006 ($^7Be$ criterion) and 1978 to 2011 (RH criterion) are given. The unsmoothed values (crosses) exhibit a strong variability, caused by slight shifts of the maxima and minima over

the very long period. It is important to consider that some of the monthly counts occur for multiple years.

The coloured lines connect the averages of the monthly intrusion counts over all the years. The lower average counts for the RH criterion agrees with the result of Trickl et al. (2010): The RH criterion yields a lower summer minimum than the $^7Be$ criterion (Fig. 13). This could be due to the wet bias of the RH minimum in the warm season (Figs. 6 and 10). In addition, a less complete overlap of the dry layer with the summit due to a higher

boundary layer must be taken into consideration.



The seasonal cycle for the entire period is influenced by an occasionally aperiodic behaviour of the monthly counts before 1995. For the ⁷Be criterion the July value is influenced by exceptionally high counts between 9 and 12 in 1983, 1990 and 1992, which disappears if these three value are removed from averaging.

Maxima and minima of the monthly counts (determined over all seasons) also do not indicate trends. For the ⁷Be criterion we obtain an average of the maximum of $8.79 \pm 1.65$ and of the minimum of $0.96 \pm 0.87$ during the entire period (1978 to 2006). The minimum values are slightly lifted due to a rise to 2 to 3 between 2000 and 2004. The minima do not necessarily occur exclusively during the warm season during the early phase. For the RH criterion the average extrema are $8.00 \pm 2.13$ and $0.38 \pm 0.54$, respectively, which conforms the absence of a trend.

The average duration of intrusions seems to be almost free of trend, which is, however, masked by variations, in particular for the RH criterion. The average duration of the intrusions for the ⁷Be criterion in a given year maximizes in winter (25 h $\pm$ 5 h around 1980 and 30 h $\pm$ 8 h around 2005), and stays at 12 h $\pm$ 4 h in summer. A winter maximum of the average duration with 40 to 60 h (⁷Be criterion) or 45 h (RH criterion) occurred between 1990 and 1995. This maximum does not appear in the corresponding annual average.

**5.2 Ozone**

In Fig. 14 we show the monthly and annual averages of ozone in direct intrusions for the RH criterion and the ±12-month arithmetic averages for both criteria. There is a pronounced positive trend of the annual peak STT ozone. The annual averages exhibit an almost periodic variation with a period of roughly seven years. The relative increase of average STT ozone from 1978 to 2011 exceeds that for the intrusion count (Sect. 5.1). This means that the average amount of ozone transported in individual intrusions also increased over the years.

The seasonal cycle of the monthly averages is somewhat clearer in structure. This allows us to present in Fig. 15 the ozone averages in the direct intrusions for January – February and June – July. The winter maxima for the two filter criteria do not differ too much. However, the summer minima for the ⁷Be are higher by roughly 50 % than those for the RH criterion due the wet bias for low RH. The equations for the four regression lines are (in ppb)

$-9.2760(1.81)\times10^2 + y\times4.7133\ (0.91)\times10^{-1}$, January – February, ⁷Be criterion,
$-1.0826(0.86)\times10^2 + y\times5.5872\ (4.30)\times10^{-2}$, June – July, ⁷Be criterion,
$-9.1827(1.94)\times10^2 + y\times4.6679\ (0.97)\times10^{-1}$, January – February, RH criterion,
$-3.8579(5.88)\times10^1 + y\times2.0444\ (2.95)\times10^{-2}$, June – July, RH criterion,

y being the year and the numbers in brackets the respective standard deviations. The strong outlier of the minima in 1983 can be understood by inspection of Fig. 12. In order to guide the eyes, we also fitted a third-order polynomial to the winter data for the RH criterion. The four parameters are

$P_0 = 7.3446\times10^6,\ P_1 = -1.1064\times10^4,\ P_2 = 5.555471,\ P_3 = -9.2976\times10^{-4}$

The relative standard deviations of the four parameters are as high as about 0.45 each. This reflects the strong year-to-year variability of the data.

It is obvious that the increase in STT ozone took mostly place during the cold season. This observation will be further discussed in Sect. 6. The new analysis for the second half of the 1990s exceeds, during the cold season,





the values of the FLEXPART analysis for the years 1995 to 1999 presented by Trickl et al. (2010) in their Fig. 1. We tentatively ascribe this fact to the excessive mixing scheme in the model (Trickl et al., 2014).

Although high-accuracy ozone data do not exist before 1978 we attempted to estimate the situation back to 1970 by assuming a constant average ozone mixing ratio (Fig. 16). This assumption is justified by the results for the nearby Hohenpeißenberg (distance: about 41 km) sonde measurements for 1970 to 1977, evaluated by Claude et al. (2001) for 700 mbar (about 3000 m). Claude et al. (2001) publish an almost constant average mixing ratio which justifies our choice. However, the Hohenpeißenberg mixing ratio before 1978 is about 41 ppb. This value

is higher than expected from the extrapolation of the Zugspitze measurements to earlier years, estimated as 36.25 ppb.

    As mentioned in the introduction the measurements of Fabian and Pruchniewicz (1977) yield somewhat low ozone mixing ratios. In Fig. 16 we display the monthly mean values for 1970 to 1975 graphically reconstructed from Fig. 5 of their paper as crosses.

Figures 16 and 17 (averaged values) also show curves for the indirect stratospheric contribution calculated with Eq. 2, for assuming 80 % and 90 % of the $^7$Be being produced in the stratosphere. The annual averages are listed in Table 1. The values for total STT ozone confirm rather well the analysis by Scheel in 2005 (Fig. 1 of Trickl et al., 2020a), considering that he assumed a stratospheric contribution of just 66.7%, i.e., the global average. The overall stratospheric contribution derived now is 12 or 14 ppb in the 1970s and 19 or 24 ppb around 2005, for the

80-% or 90-% threshold, respectively. The series for the direct intrusions in the 1970s nicely extends that for the years after 1978. The rise of $^7$Be in the early 1970s is not reproduced by the STT ozone. The difference of the total ozone mixing ratio and the stratospheric contributions is an estimate of the tropospheric burden. Most importantly, the tropospheric contribution, calculated as the difference of the full annual average mixing ratio and the estimated total stratospheric contribution, does not exhibit a positive trend after 1990, the period of

improving air quality. In any case, the ozone trend after 1990 is not negative as one could expect from the reduction in European emissions (see Introduction and Sect. 5.3).

    Despite the good performance of the results for the direct intrusions before 1978, the analysis for the indirect stratospheric contributions required a slight adjustment. The analysis showed that the influence of the seasonal cycle cannot be neglected for estimating the indirect stratospheric contribution. We, thus, added an artificial

sinusoidal seasonal cycle with amplitude 8 ppb (Fig. 16). We also realized that the "indirect" values for 1970 and 1971 were 2 to 3 ppb higher than those obtained from a sliding 12-month average for these two years. We corrected for this bias and also modified the unrealistic $O_3/^7$Be calibration factor that did not agree with the (rather constant) factor for 1978 to 2006.

    In order to extend the analysis beyond 2005 or 2011, respectively, we filtered the UFS values. No $^7$Be

measurements have been performed at UFS. Here, we just apply the RH criterion. In Fig. 18 we present the results for direct intrusions for the years 2002 to 2020. We also include the smoothed 1978-2011 results for the summit. There is an obvious difference between the two stations between 2002 and 2011. Because of the small altitude difference of 0.3 km this difference looks rather high which suggests further discussion (Sect. 6). There is, at least, some similarity in seasonal cycle. However, the UFS results look somewhat unrealistic before 2006

and we speculate on a problem with the RH measurements during the early phase.

    We multiply the UFS average by 1.35 to get some idea about the potential trend at the summit after 2011. This factor is a reasonable choice for the years 2006 to 2011. Especially before 2006 there is, still, a pronounced





difference, perhaps due to issues with the starting humidity measurements at UFS during that period. However, the important message is that the strong positive trend in stratospheric influence since 1978 seems to diminish after 2010.


The monthly averages of the measurements at both summit and UFS are also displayed in Fig. 18, together with the sliding ±12-month averages. As already evaluated by Scheel and Ries with monthly mean values from April 2002 to June 2008 the ozone at ZSF was on average 0.82 ppb lower compared to Zugspitze summit (Fig.1 of Zellweger et al., 2011). The difference is almost outside the uncertainty level for both sides. It is reasonable that


this difference is caused by a lower stratospheric influence at UFS. The slightly negative trend in Fig. 1 during the first decade of the new millennium comes to an end in 2010. Later on, STT grows again.

In Fig. 19 we evaluate the amplitude of the seasonal cycle of the overall Zugspitze ozone. The 12-month averages were subtracted from the monthly averages. There is an obvious decrease of the amplitude of the seasonal cycle since the late 1980s. We applied linear least-squares fits to the annual maxima and minima which


resulted in

$2.04667(0.63) \times 10^2 - y \times 9.7377(3.16) \times 10^{-2}$ (maxima, in ppb),

$-2.23489(0.27) \times 10^2 + y \times 1.0731(0.14) \times 10^{-1}$ (minima, in ppb).

Again, y is the year and the number in brackets are the standard deviations. In order to obtain a reasonable result, we enhanced the *a-priori* error bars of a few obvious outliers in the input data (such as the dry summer of 2003).


For the period 1988 to 2021 we obtain a relative amplitude decrease by 29 % and 35 %, respectively. This means a considerable reduction in air pollution at this near-background site.

**5.3 Carbon monoxide**

Trickl et al. (2014) show in their Fig. 17 the behaviour of the annual average of Zugspitze carbon monoxide for air inside and outside intrusion layers. The analysis was now repeated for the modified filter criteria. Figure 20


shows the results for the RH criterion for 1990 to 2011, here for the average monthly contributions. The slightly positive obtained by Scheel (Trickl et al., 2014) for CO in direct intrusions is confirmed in the revised analysis. Also the negative trend for the complementary data is confirmed. From 1990 to 2011 the averaged CO outside direct intrusions dropped from about 127 ppb to about 93 ppb. During the early 1990s the amplitude of the seasonal cycle was clearly higher than later, in agreement with the reduction of the European air pollution during


that decade.

We also analysed the UFS CO data in the same way. Since the CO data are preliminary for some years, we do not show the results here. The decrease of the averaged corrected complementary mixing ratios intensifies after 2011. By the end of 2020 a roughly estimated 72 ppb were reached, i.e., 56 % of the highest value in 1990. This means a substantial improvement of the tropospheric air quality.


By contrast, the monthly-mean CO attributed to direct intrusions stays rather constant after 2011, at about 17 ppb. Thus, the slight rise seen in the summit data from 1990 to 2011 does not continue, similar to the behaviour found for ozone. As earlier (Trickl et al., 2014) we speculate on an Asian contribution in the tropopause region, fed by the frequent off-shore warm conveyor belts over the western Pacific (Stohl, 2001). This contribution could lead to the growth of carbon monoxide. In addition, the roles of biomass burning (e.g., Fromm et al., 2010)


or air traffic must be considered.





### 6 Discussion and Conclusions

Despite substantial uncertainties we can conclude that the contribution of STT to the ozone in the lower free troposphere above the Northern Alps is rather large. The direct portion, related to descent short enough to allow identification, exceeds that in earlier work (Elbern et al, 1997; Stohl et al., 2000) as indicated in the analyses by
Trickl et al. (2010; 2020a). We successfully estimated the indirect portion, not accessible to data filtering, from the $^{7}$Be measurements. It seems that the total contribution of stratospheric ozone at the Zugspitze summit is a rather robust quantity whereas a higher uncertainty exists for the direct one. The total contribution just slightly exceeds that from the analysis of Scheel in 2005 (Trickl et al., 2020a), although we assume a larger stratospheric component of the isotope. In the earlier analysis there was less ozone from direct intrusions because of a high
$^{7}$Be threshold (85th percentile) and the indirect contribution was rather high. Now, we find a much higher direct, but a lower indirect component.

The uncertainty remains considerable, since we do not know the distribution of ozone and beryllium in the stratosphere of the source region, their modification during the transport and the radioactive decay of $^{7}$Be. $^{7}$Be is believed to be attached to aerosols that can undergo scavenging during particularly long transport (Gerasopoulos
et al., 2001; Zanis et al., 1999). From the obtained missing negative trend of the tropospheric ozone component we judge that the obtained stratospheric ozone component is more likely a conservative estimate.

We have been unable to assess the degree of overlap of the intrusion layer with the stations, which means an uncertainty in a quantification of the tropospheric admixture in the stratospheric layer. This admixture prevails next to the edges of the layers. Fortunately, incomplete overlap prevails in summer where just a few intrusion
cases are found per month at the summit.

Also the calibration of the indirect ozone component via $^{7}$Be is a source of uncertainty. However, since the average ozone does not vary much this uncertainty is presumably not very high.

The positive trend for ozone of stratospheric origin diminishes after 2010. Apart from oscillations (that could explain the negative trend in Fig. 1 before 2010) it stayed slightly positive, in agreement with findings at Uccle
(Van Malderen et al., 2021) where a positive STT trend was found until 2017. In other regions of the world also positive ozone trends have been observed (e.g., Cooper et al., 2020). It is interesting to see that the trend change follows the change in solar activity: Is there a related change in atmospheric dynamics?

The increase in stratospheric ozone observed at the Zugspitze summit predominantly occurs in winter. We found that neither the intrusion rate nor the duration of intrusions changed in a comparable manner during the long
period of observation. Claude (2003) found an increase in lower-stratospheric ozone over the Hohenpeißenberg station of DWD (distance from the Zugspitze summit: 41 km), which could explain some of the observed increase if the same was the case over the Arctic source regions. The Hohenpeißenberg increase at 11 km altitude from 1967 to 2002 is of the order of 10 % per decade. However, if rising winter-time ozone above the tropopause were the sole reason: Why did also $^{7}$Be rise?

The intrusions emerge from the lowest edge of the stratosphere (Trickl et al., 2014; 2016). As a consequence, we can also take into consideration that increasingly wider layers have separated from the range just above the tropopause where the ozone mixing ratio steeply rises and a small increase in layer width can have an enormous effect. Perhaps this reflects the growing atmospheric dynamics in the warming climate. It will be interesting to see if, with the solar activity reversing, this will come to an end (see below). A break-off of wider lower-





stratospheric layers was concluded for the warm season (Trickl et al., 2020a), but the penetration of stratospheric intrusions into the lower free troposphere in summer is limited.

The summer minimum of the monthly ozone averages due to STT is of the order of 2 ppb, the 2005 winter contribution being roughly nine times higher. FLEXPART model analyses by A. Stohl for 1995 to 1999 show a similar winter-summer contrast for downward transport times of about ten days and less (Trickl et al., 2010). The

contrast is less pronounced for the higher-lying Jungfraujoch station in Switzerland, indicating that the summer minimum might be caused by a reduced penetration of the intrusions into the troposphere during the warm season. Looking at the free troposphere as a whole the summer minimum disappears and a very high occurrence of intrusions was reported (e.g., Beekmann et al., 1997; Dibb et al. 2003; Trickl et al., 2020a).

The level of carbon monoxide in intrusions reflects the mixing ratio just above the tropopause. This level, as

conclude from the Zugspitze data, does not exhibit a major change, apart from perhaps a slight increase until 2004. Trickl et al. (2014) speculated on an Asian influence in the tropopause region, possibly fed by warm-conveyor-belt activity over the western Pacific (Stohl, 2001). Just the tropospheric CO decreases, even to roughly 56 % of the value in 1990 towards the end. The tropospheric ozone component estimated in our analysis (Fig. 17) does not decrease in a similar way, which confirms the idea of a strong stratospheric fraction.

Certainly, improved modelling will be needed in addition to quantify STT. So far, Eulerian models have had difficulties in reproducing the strong ozone rise at the Alpine sites (e.g., Parrish et al., 2014; Staehelin et al., 2017). The calculated ozone rise reported by Parrish et al. (2014) and Staehelin et al. (2017) ends almost 20 years earlier than the observed one. In most commonly used Eulerian models the spatial resolution is too low to reproduce deep STT (Roelofs et al., 2003; Trickl et al., 2010; Rastigejev et al., 2010; Eastman and Jacob, 2017),

and free-tropospheric mixing must be reduced (Trickl et al., 2014). Due to the limited free-tropospheric mixing Lagrangian approaches look promising since they have a better chance to capture thin layers.

In any case, an extension of transport modelling to 20 days and more is desirable, implying high spatial and temporal resolution. Our studies (e.g., Trickl et al., 2020a) have revealed that, with growing altitude, the transport pattern of the intrusions affecting the free troposphere over the Northern Alps is increasingly

characterized by slow descent from Canada, Alaska and Siberia (Type 6 as defined by Trickl et al. (2010) over more than ten days. The trajectories frequently exhibit horizontally wavelike transport paths, but mostly without strong vertical variation. This kind of long-range descent, its underlying dynamics and its influence on the STT budget call for a meteorological explanation. It would also be interesting to determine how much an extension of the transport calculations to at least fifteen days (as suggested by our analyses) would change the STT budget

with respect to earlier work.

The great advantage of the *in-situ* measurements is their continuous operation which excludes a fair-weather bias. In addition, for the Zugspitze summit information is available from the [7]Be measurements. All this makes the data filtering a valuable approach. However, such a filtering effort must, in principle, also account for the source conditions: The atmosphere in the tropopause region was estimated to be mixture of about 50%

stratospheric and tropospheric air each (Shapiro, 1980; Vogel et al., 2011). The stratospheric portion of the descending air mass can vary significantly, also depending on the stratospheric residence time (Reiter et al., 1975). However, for our considerations we name an air mass stratospheric once it has resided in the stratosphere at least for a short period of time. All this calls for refined modelling efforts.



The growth of stratospheric influence at the Zugspitze site and elsewhere indicates a drying of the free troposphere. This can be directly seen in the RH results in Figs. 7 and 9. More generally, Paltridge et al. (2009) have determined a negative humidity trend in the global free troposphere over four decades from analysed sonde data (via NCEP re-analysis). The negative trend maximizes in the upper troposphere where we found also the maximum of STT (Trickl et al., 2020a). Based on our results we conclude that STT could contribute to this negative humidity trend. STT occurs in many regions, mostly in the latitudinal bands around the jet streams, but

also elsewhere. Is there a reaction of vertical exchange to the changing climate?

Tropospheric drying would be primarily expected from precipitation. However, for Germany the German Weather Service (DWD) determined almost constant precipitation since at least 1950 (https://www.dwd.de/DE/leistungen/zeitreihen/zeitreihen.html?nn=480164, under key words "Niederschlag" (precipitation)and "Jahr" (year)), with year-to-year variations of up to about ±25 %. Thus, the role of STT on the

tropospheric drying until the beginning of the new century could be even rather important. The drying of the free troposphere contradicts the expectations from climate modelling. However, as mentioned above, deep STT is likely to be missed by climate models to a major extent because of their coarse grids.

## 7 Data availability

The data used in this paper can be obtained on request from the authors (thomas@trickl.de;

hannes.vogelmann@kit.de; cedric.couret@uba.de; ludwig.ries@gawstat.de). We follow the strict conventions in renowned international networks. The hourly Zugspitze and UFS ozone values are available in the World Data Center for Reactive Gases (WDCRG: https://ebas.nilu.no/) and the TOAR data base (Schultz et al., 2017). Most of the UFS CO data are stored by the World Data Center for Greenhouse Gases in Tokyo (WDCGG: https://gaw.kishou.go.jp/). Relative humidity data for both the summit and UFS are freely available from DWD

(see Sect. 2.2.3).

## 8 Author statement

TT interpreted the observations and prepared most of the manuscript, based on studies interrupted by the death of H. E. Scheel, and assisted by the co-authors. TT and HV carried out the lidar measurements. CC carried out GAW measurements at UFS and provided the data for the most recent years, LR led the GAW activities of UBA

at UFS until 2019 and contributed details on the GAW measurements there.

## 9 Competing interests

The authors declare that they have no conflict of interest.

## Acknowledgements

The authors thank the late Reinhold Reiter, as well as Wolfgang Seiler and Hans Peter Schmid for their support

over that many decades. They are indebted to the late Hans-Eckhart Scheel, who can no longer co-author this paper, and the numerous coworkers of IFU (IMK-IFU) who have contributed to maintaining the data acquisition at the *in-situ* stations of the institute, in part listed in the publications cited. Samuel Oltmans kindly returned Zugspitze ozone data for five years not yet archived. The $^7$Be measurements were carried out over an extended





period of time by the late Hans-Joachim Kanter, as well in the final phase by Alexander Rockmann. At UFS,
Ralf Sohmer carried out the ozone measurements over many years and Steffen Knabe the CO measurements.
Johann Siemens provided information on the *in-situ* humidity instrumentation of the German Weather Service
(DWD), Birgit Wegstein and Thomas Elste the RH data of DWD for the summit and UFS, respectively. Alfred
Neururer sent sonde data for Innsbruck. The authors acknowledge the great support by the UFS team. The
measurements at Wank and Zugspitze have been funded by numerous agencies, in particular the German
Bundesministerium für Bildung and Forschung within EUROTRAC, ATMOFAST and several projects, UBA,
the European Union within VOTALP 1 and 2 (Vertical Ozone transport in the Alps, Wotawa and Kromp-Kolb,
2000) and STACCATO (Influence of Stratosphere-Troposphere Exchange in a Changing Climate on
Atmospheric Transport and Oxidation Capacity, Stohl et al., 2003). The measurements at the Wank and
Zugspitze stations have contributed to EUROTRAC within the TOR (Tropospheric Ozone Research) subproject
(Kley et al., 1997). Lidar measurements contributed to TOR, EARLINET (European Aerosol Research Lidar
Network, 2003), the latter currently partly funded within the European infrastructure ACTRIS.

The service charges for this open access publication have been covered by a Research Centre of the Helmholtz
Association.



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

**Table 1.** Annual ozone averages for Zugspitze and UFS [ppb]; the asterisks denote total
ozone values estimated from extrapolation.

| Year | Direct | Indirect (80 %) | Indirect (90 %) | Total | STT total (80 %) | STT total (90 %) | Tropospheric (80 %) | Tropospheric (90 %) |
|---|---|---|---|---|---|---|---|---|
| 1970.50 | 3.96 | 9.56 | 11.65 | 36.25* | 13.52 | 15.61 | 22.73 | 20.64 |
| 1971.50 | 4.33 | 11.60 | 13.89 | 36.25* | 15.94 | 18.23 | 20.31 | 18.02 |
| 1972.50 | 3.92 | 8.64 | 10.46 | 36.25* | 12.56 | 14.38 | 23.69 | 21.87 |
| 1973.50 | 4.98 | 8.36 | 10.35 | 36.25* | 13.34 | 15.33 | 22.91 | 20.92 |
| 1974.50 | 2.03 | 10.83 | 12.80 | 36.25* | 12.86 | 14.83 | 23.39 | 21.42 |
| 1975.50 | 5.03 | 7.63 | 9.59 | 36.25* | 12.66 | 14.62 | 23.59 | 21.63 |
| 1976.50 | 3.90 | 8.78 | 10.72 | 36.25* | 12.68 | 14.61 | 23.57 | 21.64 |
| 1977.50 | 3.42 | 8.30 | 10.10 | 36.25* | 11.72 | 13.51 | 24.53 | 22.74 |
| | | | | | | | | |
| 1978.50 | 4.67 | 6.76 | 8.56 | 36.43 | 11.43 | 13.23 | 25.00 | 23.20 |
| 1979.50 | 3.19 | 9.19 | 11.15 | 37.31 | 12.37 | 14.33 | 24.93 | 22.97 |
| 1980.50 | 5.67 | 5.71 | 7.82 | 39.06 | 11.38 | 13.49 | 27.67 | 25.57 |
| 1981.50 | 4.23 | 8.24 | 10.27 | 42.39 | 12.47 | 14.50 | 29.92 | 27.89 |
| 1982.50 | 6.55 | 11.29 | 14.07 | 49.01 | 17.84 | 20.62 | 31.17 | 28.39 |
| 1983.50 | 7.37 | 10.65 | 13.31 | 46.29 | 18.02 | 20.67 | 28.27 | 25.62 |
| 1984.50 | 5.11 | 10.10 | 12.52 | 44.58 | 15.20 | 17.63 | 29.38 | 26.95 |
| 1985.50 | 4.28 | 11.74 | 14.24 | 44.12 | 16.01 | 18.52 | 28.11 | 25.60 |
| 1986.50 | 6.85 | 10.28 | 12.95 | 47.74 | 17.13 | 19.80 | 30.61 | 27.94 |
| 1987.50 | 4.97 | 10.93 | 13.52 | 47.21 | 15.90 | 18.49 | 31.32 | 28.72 |
| 1988.50 | 5.09 | 7.93 | 9.97 | 46.90 | 13.02 | 15.07 | 33.88 | 31.84 |
| 1989.50 | 8.19 | 9.60 | 12.43 | 48.75 | 17.79 | 20.62 | 30.95 | 28.13 |
| 1990.50 | 7.13 | 11.03 | 13.90 | 50.73 | 18.16 | 21.03 | 32.57 | 29.71 |
| 1991.50 | 8.14 | 9.75 | 12.48 | 48.26 | 17.89 | 20.62 | 30.38 | 27.64 |
| 1992.50 | 8.02 | 8.53 | 11.33 | 50.29 | 16.55 | 19.34 | 33.75 | 30.95 |
| 1993.50 | 6.81 | 10.15 | 12.94 | 48.93 | 16.96 | 19.75 | 31.97 | 29.18 |
| 1994.50 | 5.46 | 10.40 | 13.11 | 48.78 | 15.86 | 18.57 | 32.92 | 30.21 |
| 1995.50 | 5.90 | 12.03 | 14.82 | 51.53 | 17.93 | 20.72 | 33.61 | 30.81 |
| 1996.50 | 8.30 | 9.58 | 12.36 | 51.74 | 17.89 | 20.66 | 33.85 | 31.08 |
| 1997.50 | 8.62 | 10.24 | 13.10 | 50.49 | 18.86 | 21.72 | 31.63 | 28.77 |
| 1998.50 | 9.66 | 8.95 | 11.87 | 52.47 | 18.61 | 21.54 | 33.86 | 30.93 |
| 1999.50 | 8.31 | 9.80 | 12.93 | 51.60 | 18.12 | 21.24 | 33.49 | 30.36 |
| 2000.50 | 8.27 | 8.64 | 11.53 | 49.87 | 16.92 | 19.80 | 32.96 | 30.07 |
| 2001.50 | 8.07 | 9.52 | 12.33 | 51.09 | 17.60 | 20.41 | 33.49 | 30.68 |
| 2002.50 | 9.39 | 7.62 | 10.91 | 51.44 | 17.01 | 20.31 | 34.42 | 31.13 |
| 2003.50 | 13.16 | 10.19 | 13.52 | 54.66 | 23.35 | 26.68 | 31.31 | 27.98 |
| 2004.50 | 8.45 | 10.71 | 13.57 | 49.99 | 19.16 | 22.02 | 30.83 | 27.97 |
| 2005.50 | 9.00 | 10.28 | 13.21 | 49.93 | 19.29 | 22.22 | 30.65 | 27.72 |




**Figures:**



**Fig. 1.** Monthly mean ozone mixing ratios for the Zugspitze and Wank summits from 1978 to 2010; the black curves represent the deseasonalized values.



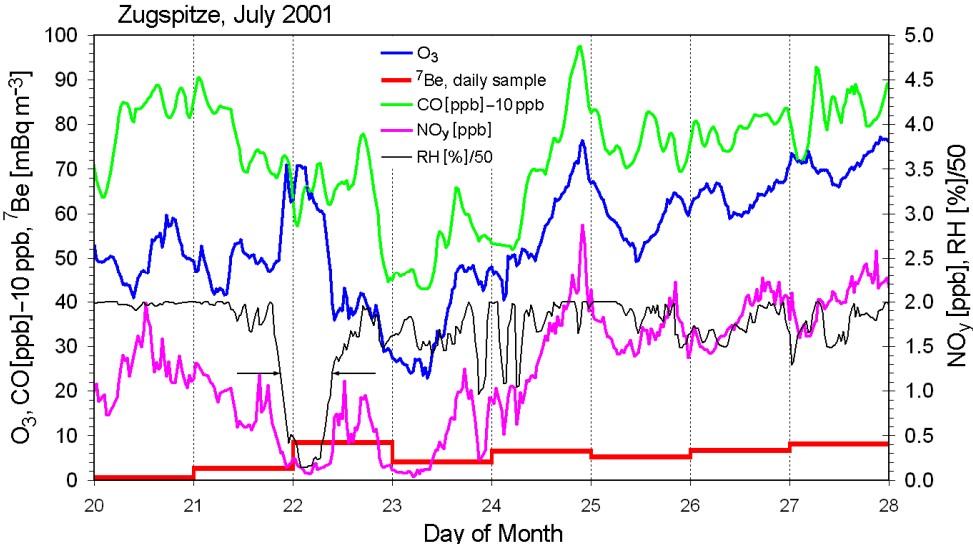

**Fig. 2.** Measurements of Ozone, $^7$Be, CO, NO$_y$ and RH at the Zugspitze summit between 20 and 27 July 2001; the 60-%-RH level during an intrusion event on 21 and 22 July is marked by two horizontal black arrows.

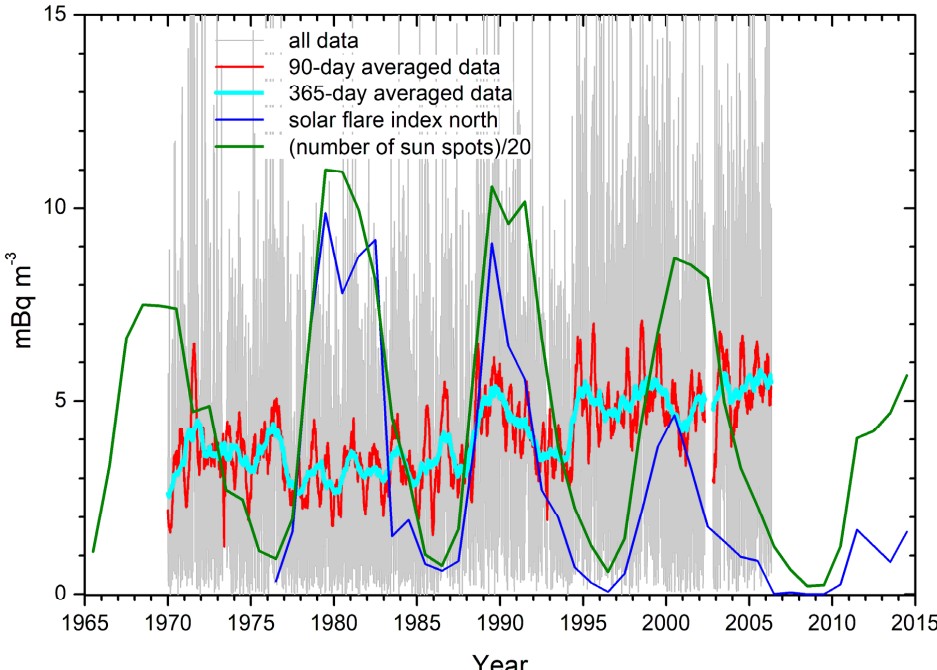

**Fig. 3.** Time series of $^7$Be from 1979 to April 2006: the gray curve represents all 24-h measurements, the red and cyan sliding 90- and 365-ay averages, respectively. In addition, we show time series of the Solar Flare Index for the northern hemisphere and the annual sun-spot count.



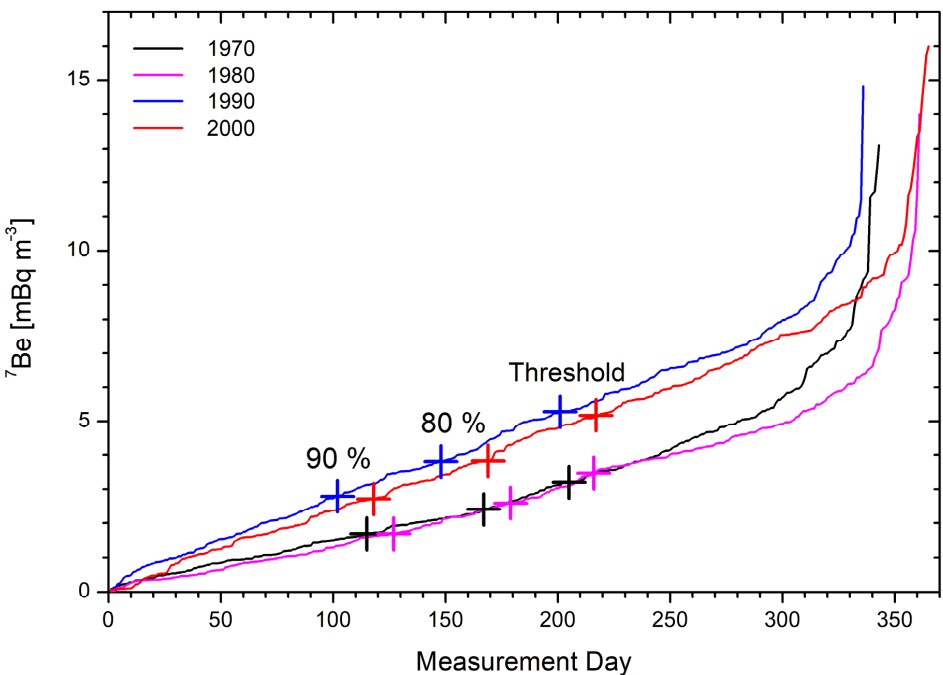

**Fig. 4.** Selected $^7$Be annual series for 1970, 1980, 1990 and 2000 with the values sorted from low to high as needed for calculating percentiles; the crosses labelled mark the points where the downward integral reaches 80 % and 90 % of the full area. In addition, the 60$^{th}$ percentile used as the threshold for the data filtering is marked.



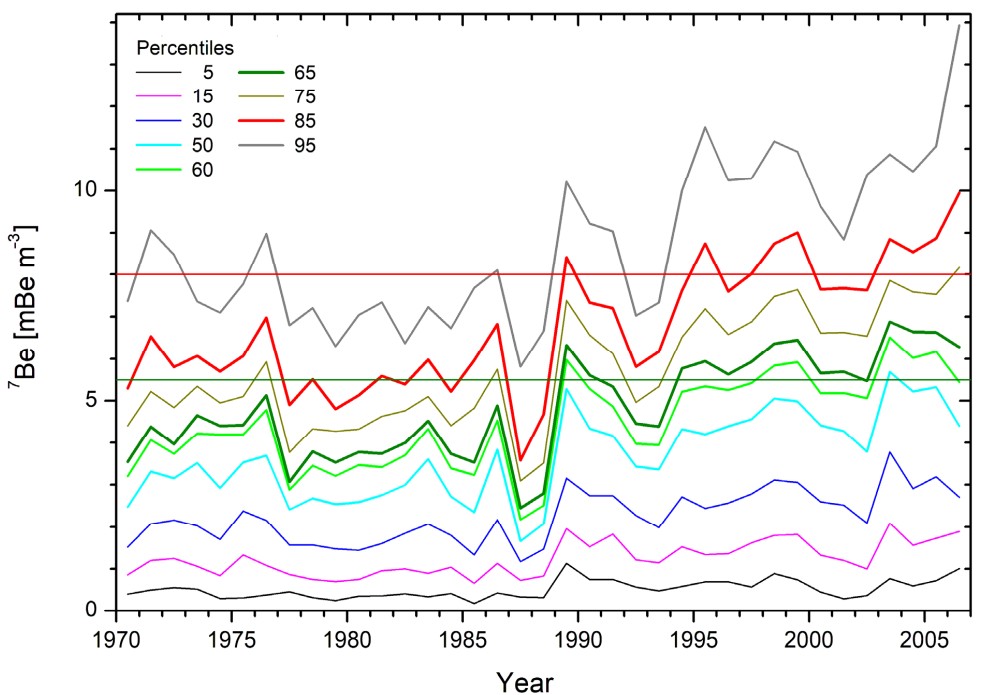

**Fig. 5:** Annual percentiles of $^{7}$Be for the entire Zugspitze measurements series from 1970 to April 2006; the horizontal lines mark the 8.0 (red) and 5.5 (oliv) mBq m$^{-3}$ thresholds explained in the text.






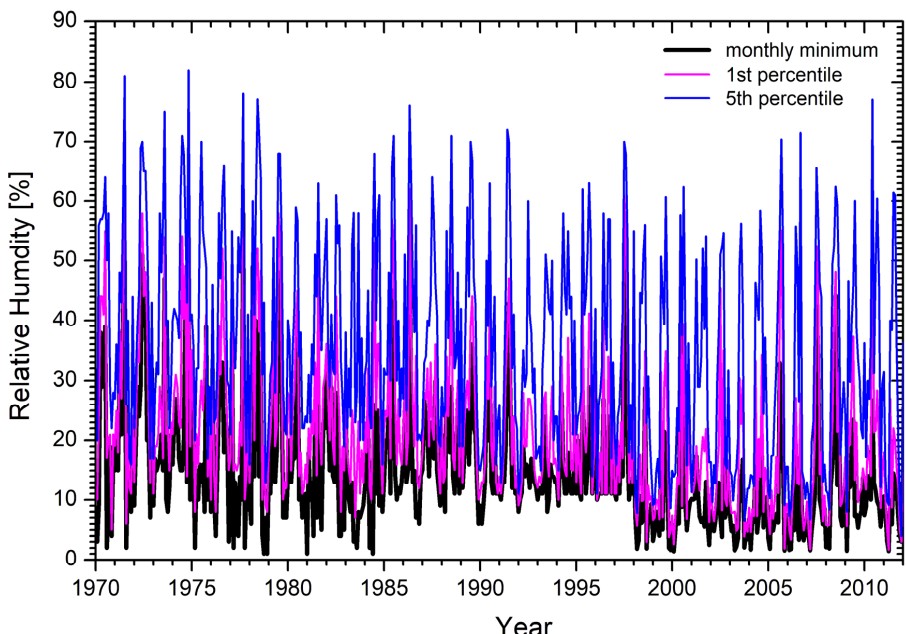

**Fig. 6.** Selected percentiles of the Zugspitze relative humidity between 1970 and 2012


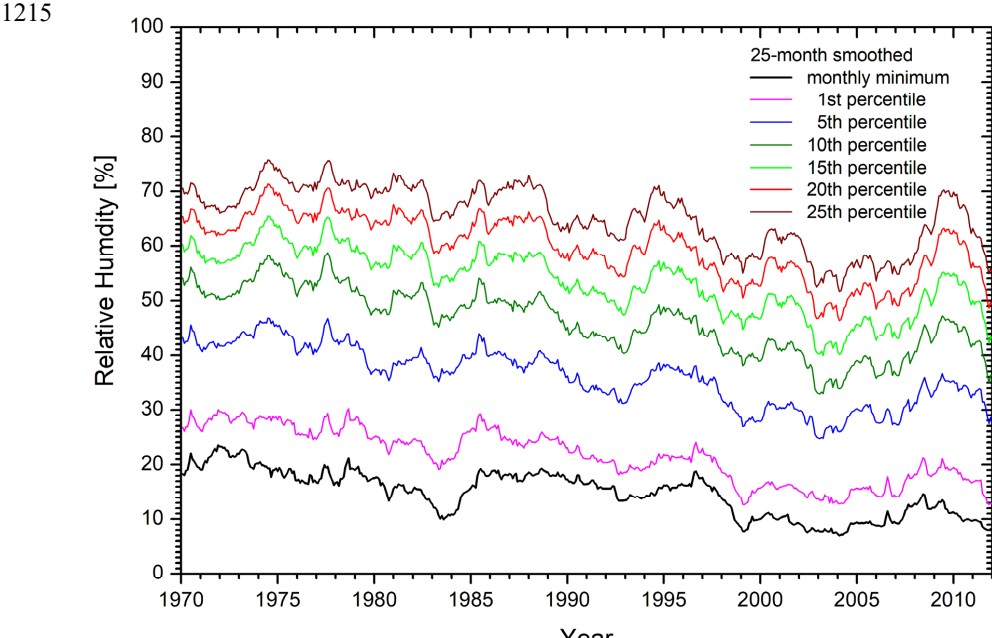

**Fig. 7.** Selected monthly percentiles of the Zugspitze relative humidity between 1970 and 2012, arithmetically averaged over ±12 months; for the lowest curves a positive offset is seen that can be explained by the use of a different sensor type during that period.



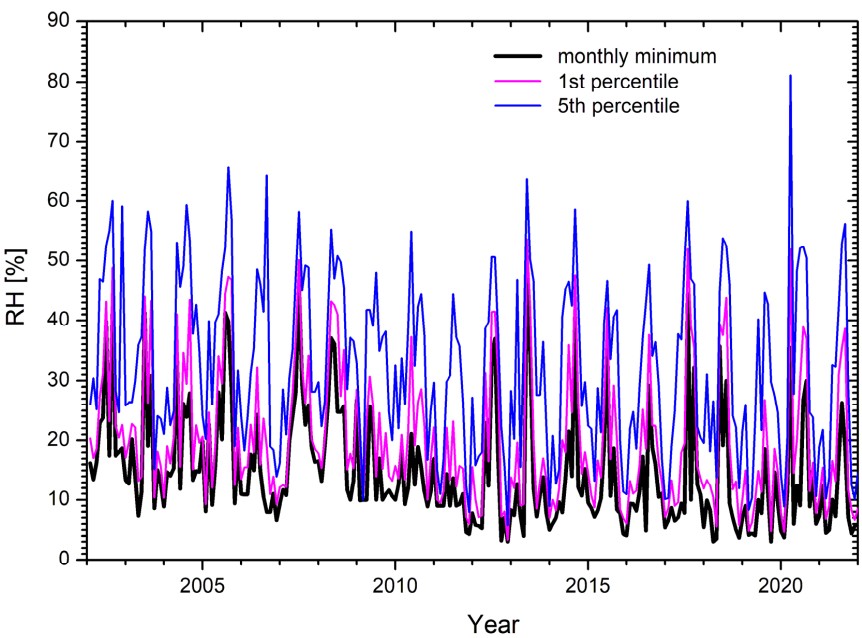

**Fig. 8.** Selected percentiles of the UFS relative humidity between 2002 and 2021

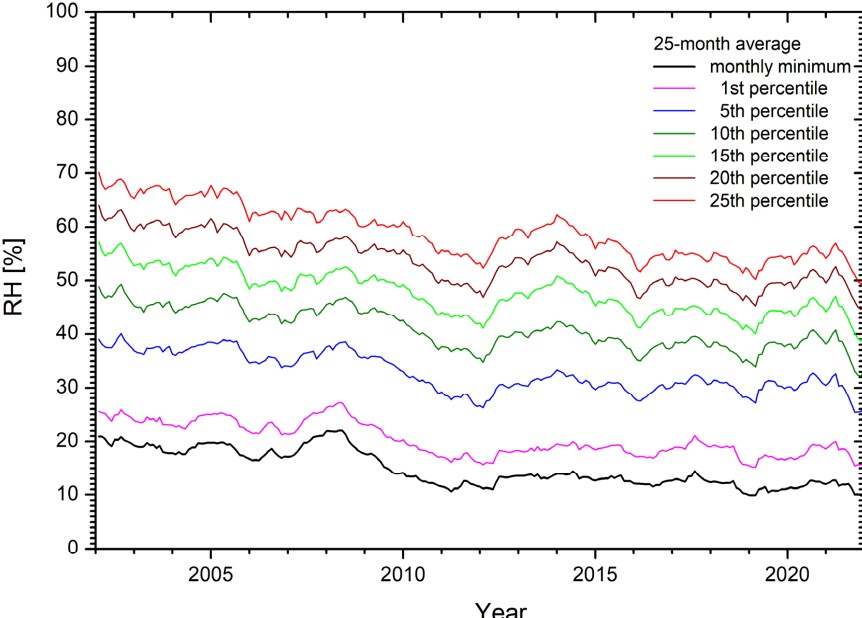

**Fig. 9.** Selected monthly percentiles of the UFS relative humidity between 2002 and 2021, arithmetically averaged over ±12 months; for the lowest curves a positive offset is seen that can be explained by the use of a different sensor type during that period.


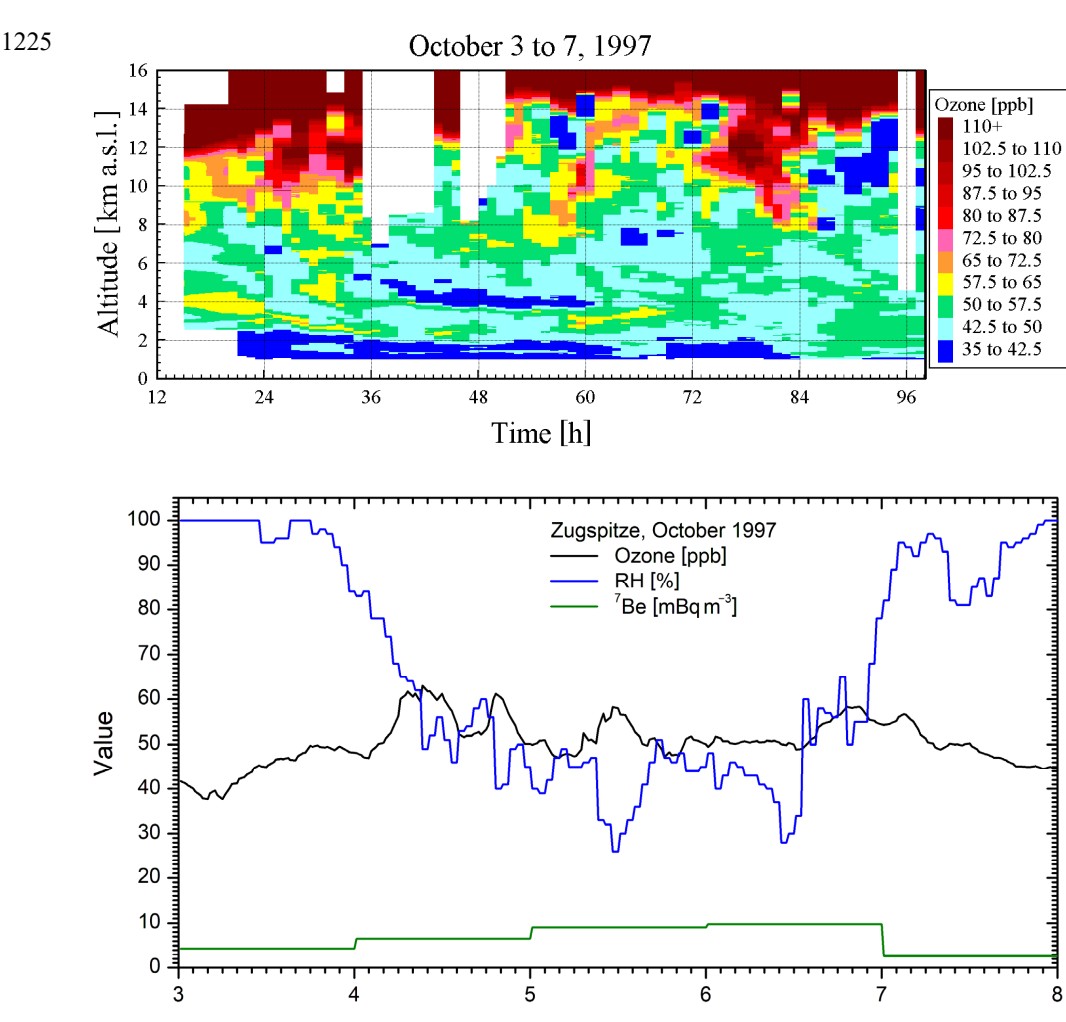

**Fig.10.** Lidar (upper panel) and Zugspitze (lower panel) measurements on 3 to 7 October 1997; the tiny ozone four peaks in the Zugspitze ozone match the crossings of the elevated-ozone layers in the lidar measurements with 3000 m after 28 h, at 60 h and after 87 h. The strongly elevated $^7$Be specific activity on 4 to 6 October suggests the presence of stratospheric air, despite the low ozone rise.






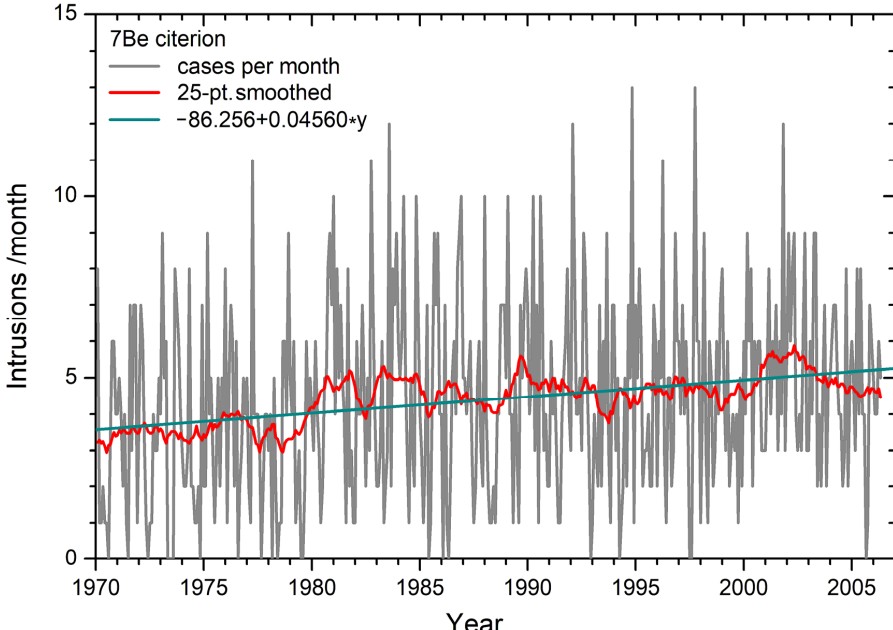

**Fig. 11.** Number of intrusions per month reaching the Zugspitze summit between 1978 and April 2006, based on the [7]Be criterion; short events (≤ 2 h) during which the criterion was fulfilled were discarded (see text).


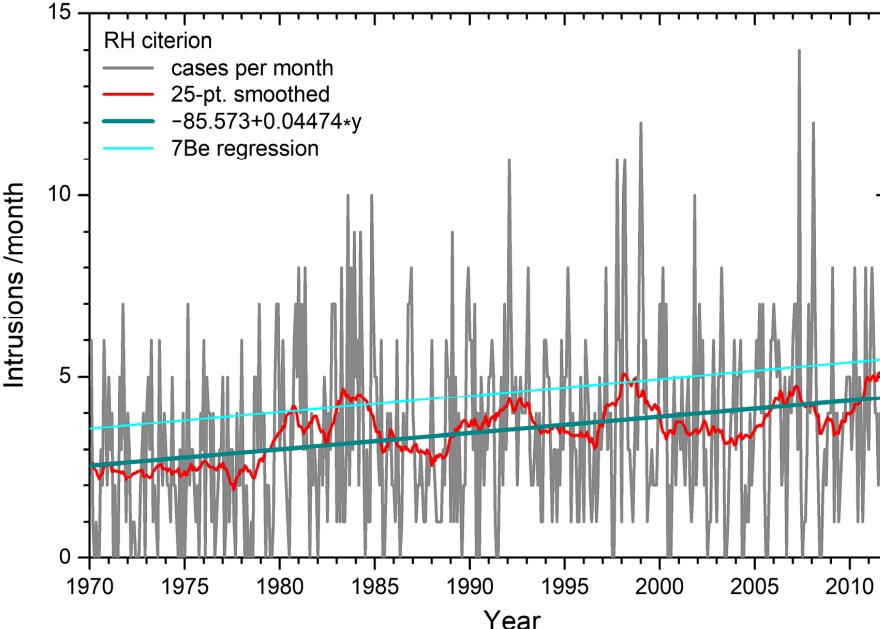

**Fig. 12.** Number of intrusions per month reaching the Zugspitze summit between 1978 and April 2006, based on the [7]Be-RH criterion; short events (≤ 2 h) during which the criterion was fulfilled were discarded (see text).





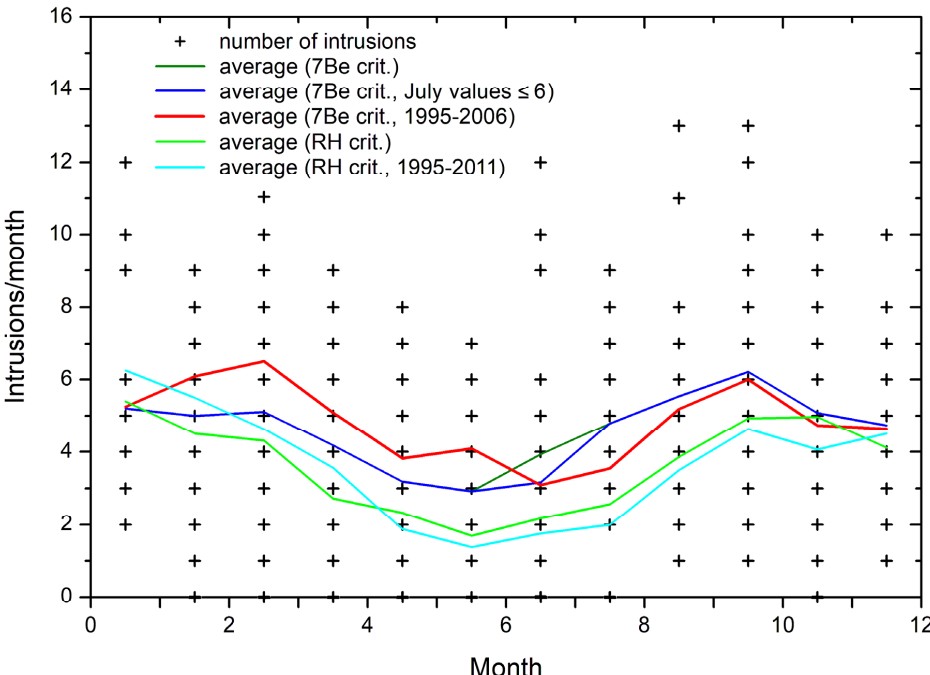

**Fig. 13.** Seasonal cycles of the monthly intrusion count for different periods; the crosses represent all monthly counts from 1978 to 2006 for the [7]Be criterion. Many of the crosses represent more than a single month over the years. The coloured lines connect the averages of the monthly counts over all years. As already evaluated by Trickl et al. (2010) the summer minimum is clearly smaller for the RH criterion than for the [7]Be criterion.

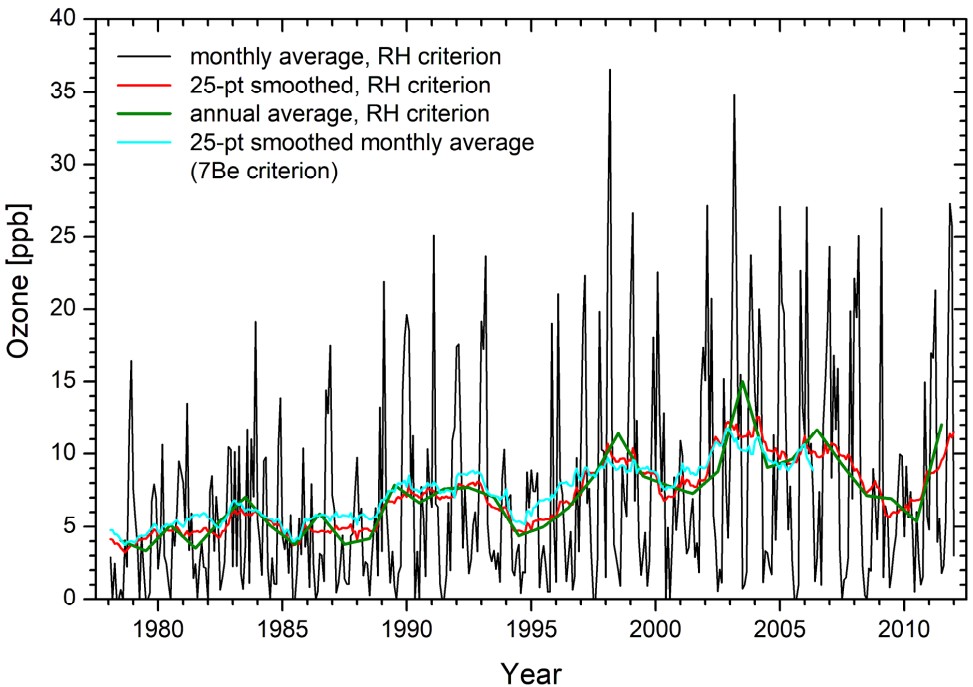

**Fig. 14.** Monthly (black) and annual averages of half-hour ozone in direct intrusions for the RH (black, red, dark green) and the $^7$Be criterion (cyan), 1978 to 2011; 25-point average means a running ±12-month arithmetic average.

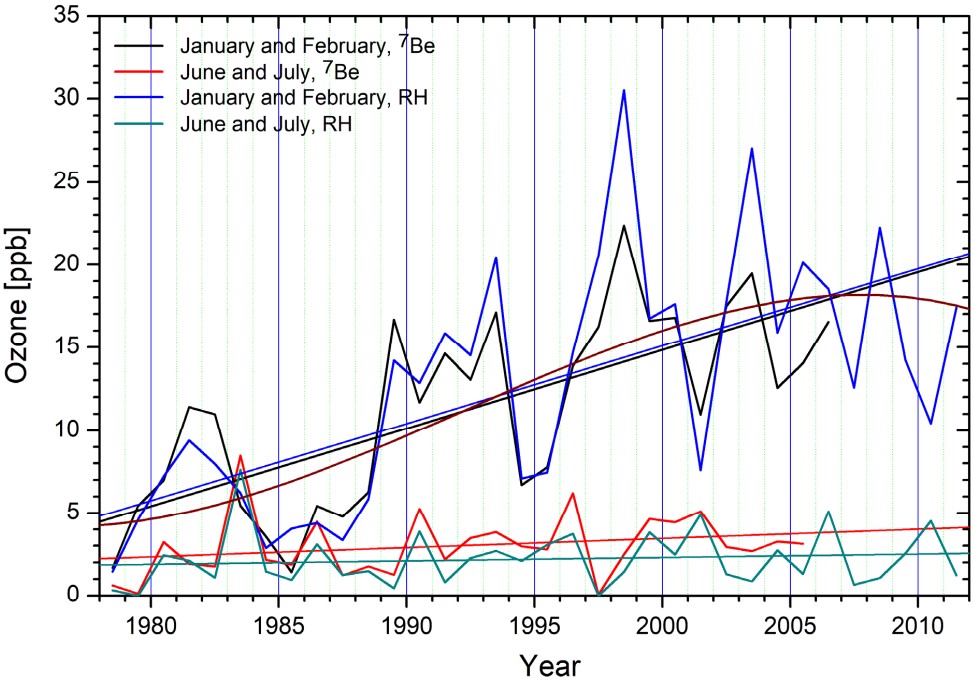

**Fig. 15.** Monthly averaged half-hour ozone in intrusions for January-February and June-July for the two filtering criteria; the results for linear regressions are shown as straight lines in the same colour as the values obtained from the analyses as well as a curved line for a third-order polynomial fitted to the winter data for the RH criterion in dark red. The year scale was shifted by 0.5 years to centre the annual averages in the middle of a given year.





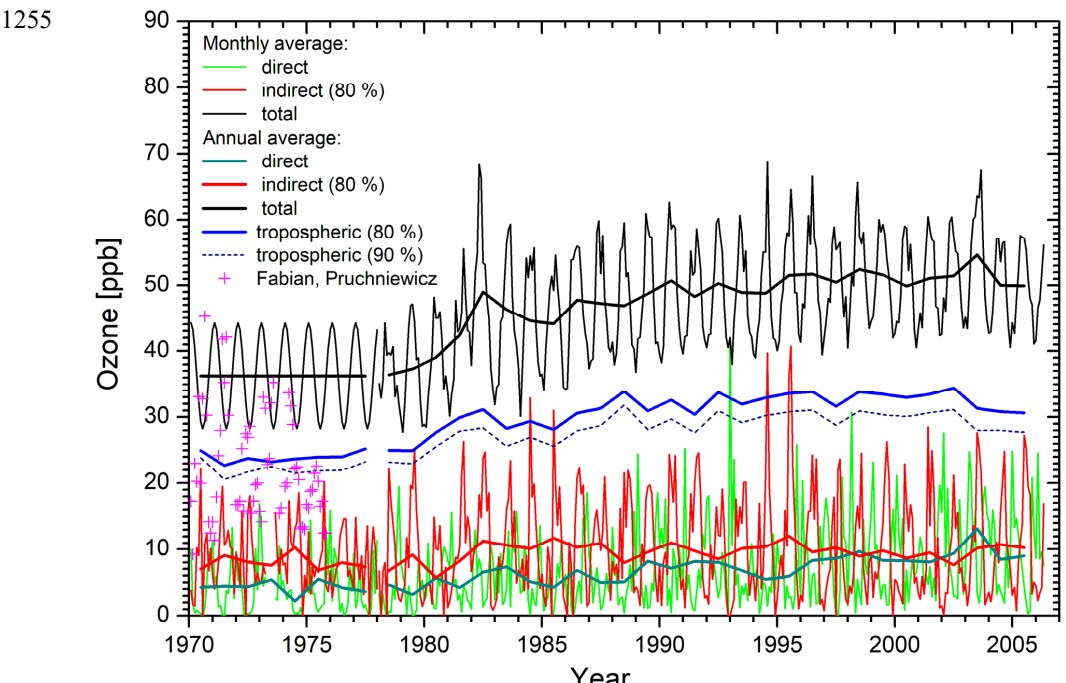

**Fig. 16.** Monthly and annual averages of ozone obtained for the ⁷Be criterion; for the period before 1978 we assume a constant mixing ratio of 36.25 ppb that matches 1978 annual ozone average. The different curves are explained in the text.






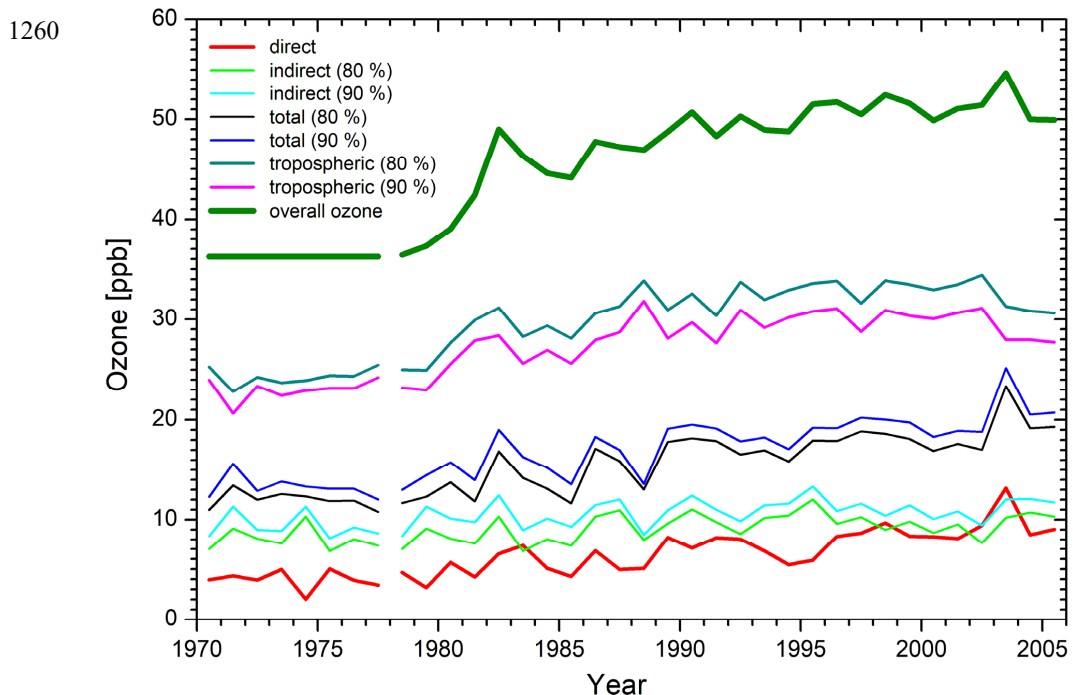

**Fig. 17.** Annual averages of ozone in intrusions identified with the [7]Be criterion; for the period before 1978 we assume a constant annual-average mixing ratio of 36.25 ppb monthly modulated as shown in Fig. 16. 36.25 ppb approximately matches the 1978 annual ozone average. The different curves are explained in the text.



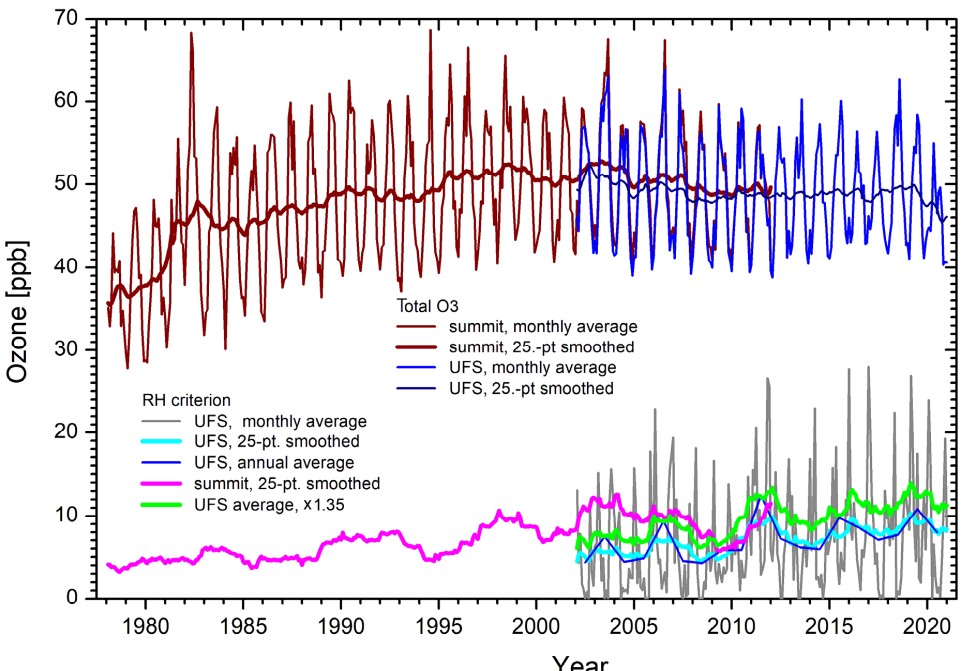

**Fig. 18.** Monthly averages of UFS half-hour ozone in direct intrusions (dark grey) and sliding ±12-month averages (cyan and magenta, respectively) for both UFS and summit, 1978 to 2020, all for the RH criterion; the smoothed UFS values are also shown multiplied by 1.35 (green) to obtain some idea about the overall STT trend. In addition, we show the monthly ozone average for the summit (dark red) and UFS (blue), also smoothed over ±12 months (25-pt.; dark red and dark blue, respectively).

This figure should be printed over two columns





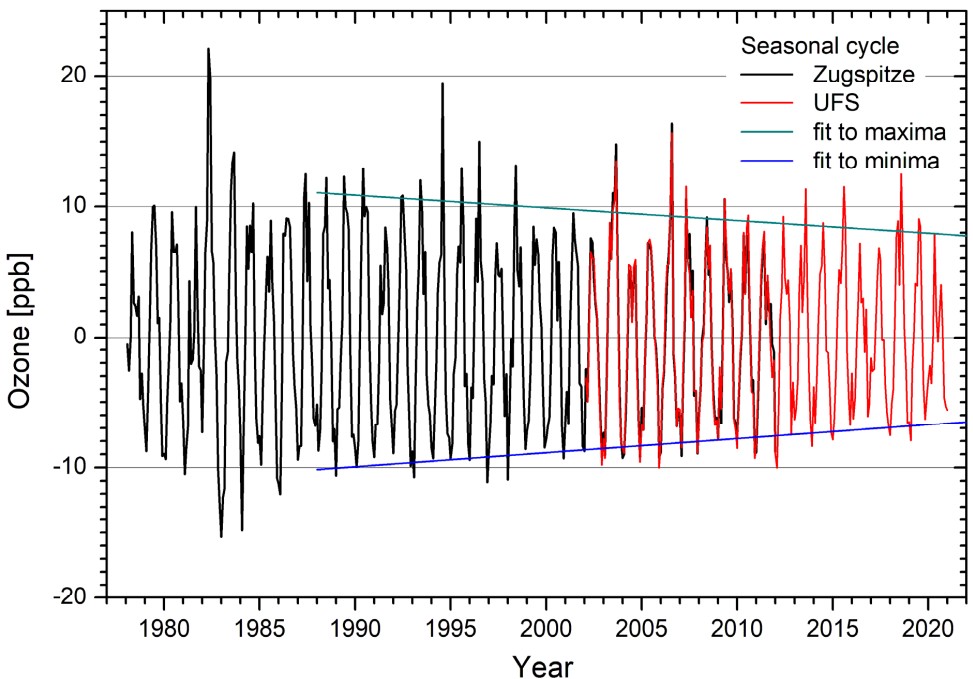

**Fig. 19.** Seasonal cycles of the monthly averages of the total half-hour ozone values for Zugspitze and UFS; the ±12-month averages are subtracted. Linear lest-squares fits to the seasonal maxima and minima are shown (1988 to 2020).


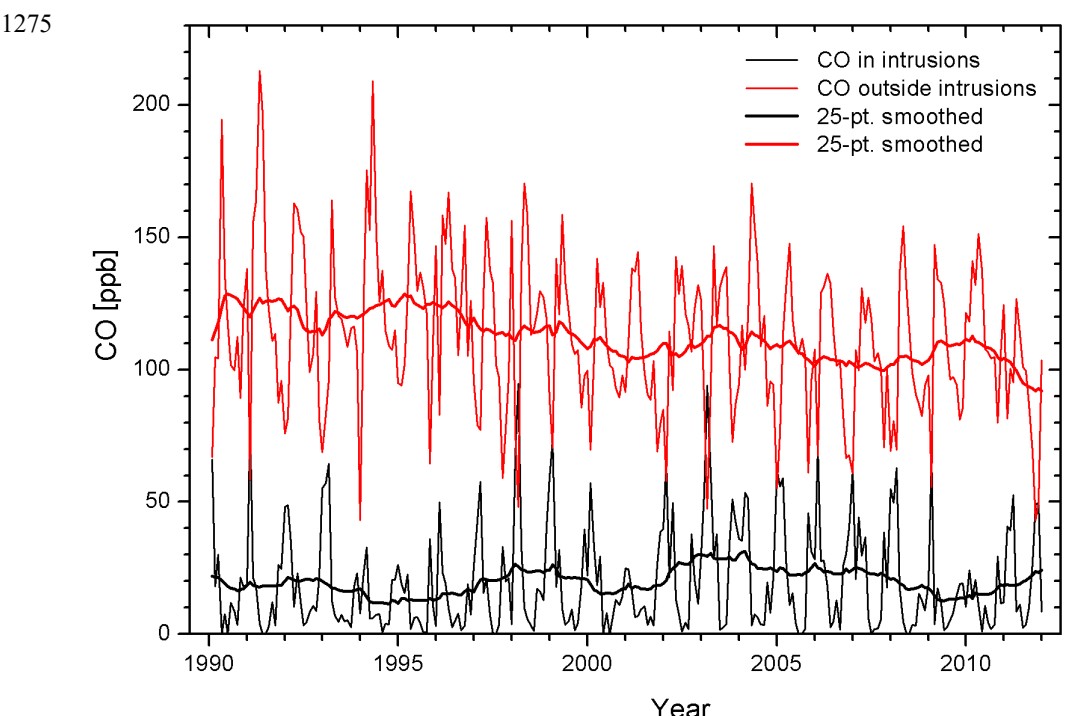

**Fig. 20.** Averaged half-hour Zugspitze CO mixing ratios for the months from 1990 to 2011, based on filtering with the RH criterion; in addition, the curves for applying sliding ±12-month averages are shown.





**Extended version:**

**7 Data availability**

The data used in this paper can be obtained on request from the authors (thomas@trickl.de; hannes.vogelmann@kit.de; cedric.couret@uba.de; ludwig.ries@gawstat.de). We follow the strict conventions in renowned international networks. The hourly Zugspitze and UFS ozone values are available in the World Data Center for Reactive Gases (WDCRG: https://ebas.nilu.no/) and the TOAR data base (Schultz et al., 2017). Most of the UFS CO data are stored by the World Data Center for Greenhouse Gases in Tokyo (WDCGG: https://gaw.kishou.go.jp/). Relative humidity data for both the summit and UFS are freely available from DWD (see Sect. 2.2.3).