# Peer review of "Zugspitze ozone 1970 – 2020: The role of stratosphere-troposphere transport"

_Atmospheric Chemistry and Physics, 2022_

## Author Comment (AC1)

**Reply to the reviews of manuscript ACP-2022-783,**

**Zugspitze ozone 1978–2020: The role of stratosphere-troposphere transport**

by Thomas Trickl, Cédric Couret, Ludwig Ries, and Hannes Vogelmann

May 22, 2023

Both reviewers recommend shortening of the manuscript and we try to follow their suggestions as far as possible. Indeed, we found almost identical information spread over the manuscript that is now shifted to the sections where it belongs to. This also allowed us to remove some text that became unnecessary. Finally, some parts were modified to improve the clearness.

In the following the reviewers´ comments are printed here in italics, the replies in normal text.

**Reply to Review 1:**

*General comments:*

*The manuscript presents an interesting and extensive analysis of more than 40 years of ozone data measured at the Zugspitze summit and the nearby Schneefernerhaus research station in the German Alps. The present manuscript builds on a series of publications by some of the authors. The manuscript reveals some interesting phenomena like that increasing stratospheric contribution to the O3 levels at Zugspitze over time. However, the paper is hard to follow as it is rather wordy, uses some vague terminology, mixes previously published results, methods and new findings, and includes an excessive number of figures. See some examples in the specific comments below.*

*To my mind, the paper is within the scope of 'Atmospheric Chemistry and Physics' but can profit from considerable shortening and removal or movement of some of the figures to the supplementary material and, thus, requires major revisions prior to its acceptance for ACP.*

As mentioned above we re-organized some of the text which allowed us to to shorten or remove some of the paragraphs. Indeed, there are pieces of information that show up parallel in different sections. In addition, we removed Figs. 8, 9 and 13. The rest is seen as necessary. A total of 17 figures is acceptable in a full-size paper. We do not favour supplementary material.

*Specific comments:*

*Line 19: it is confusing that changes between 1970 and 2003 are reported while the title of manuscript refers to the 1978 to 2020 period.*

Thank you for pointing this out! We modified the abstract for more clearness.

*Lie 21ff: "The air masses hitting the Zugspitze summit have become drier in the percentile range up to at least 25 % until 2003 ..." difficult/impossible to understand. Please rephrase.*

Indeed, the information on the percentiles is too much here and was removed.

*Lines 23-24: "... a growing width of the intrusion layer departing downward from just above the tropopause must be taken into consideration ..." difficult to understand here (and on page 20)*

We replaced "width" by "thickness". A thicker layer from above the tropopause should contain higher ozone!

*Line 25: "... perhaps a slight increase ...". Very vague wording. Is it statistically significant?*

We removed "perhaps" and the end of the sentence.

*Lines 46 ff and Fig. 1, lower panel: It is uncommon to refer to figures already in the introduction. I do not see the added value of showing the Wank data here. It does not really contribute to the discussions and conclusions drawn here.*

Indeed, one rarely finds figures in introductions, but such cases do exist. This figure is not a result of this paper and represents existing knowledge, i.e., it belongs to the introduction. It has been shown just on conferences, but has not been published not in the corresponding abstracts. Thus, it makes sense to present this figure here. The figure is special since one rarely finds neighbouring sites with high-quality atmospheric measurements at two strongly differing altitudes! It strongly supports our view.

*Lines 89-90: add reference to the "far below 1%"-statement.*

The corresponding references were listed after the following sentence, but are now shifted to the first sentence.

*Lines 121 ff.: Awkward sentence (in grammatical terms). Please rephrase.*

Thank you! Changed!

*Lines 179-179: add references or urls, according to the WCC webpage (at Empa), there was another audit in 2020.*

References added!

*Line 182: add url of the data centre.*

Added!

*Line 199: add location or url for Deuste-Steininger.*

Added!

*Line 201: add year 2020? See above.*

Added!

*Lines 217-2018: "The periods are time shifted to 7:30 to 19:00 CET and 19:30 to 7:00. This does not matter since the data filtering is carried out on a half-hour basis." Difficult to understand since the filtering is only explained below.*

We had not been able to clarify the true meaning of the units since both H. E. Scheel and H.-J. Kanter passed away, and DWD could not help us. Meanwhile, DWD provided data for 2005 that allowed us to make a comparison. This comparison clearly hardened that Dr. Scheel converted the DWD 12-h values to 24 h. Therefore, we could simplify the text.

*Line 220: "... both averages determined on a half-hour grid ..." This is confusing. What does it mean? 7Be were measured as 12h or 24h bulk samples.*

*Lines 220-221: "This means that the 12-h DWD data were calibrated to match the 24-h IFU data. It could not be clarified why this is the case." What does that mean? The DWD were scaled to match the IFU data? You cannot call this a calibration. Your last sentence is weird ... and I do not understand to what it refers. To the application of a scaling factor? If so, don't you think that it was done to homogenize the two records?*

As mentioned: the text was changed.

*Line 228-229: Something is linguistically wrong in this sentence.*

Changed!

*Lines 231-232: "... which turned out to be highly necessary near the temporal boundary (end) of our analysis." Weird wording. How about "... which were useful to apply to the filters (see below) to the whole dataset."*

Deleted!

*Chapter 3 is can be considerably shortened as it largely repeats previous findings. However, it also mixes methods, previous findings and new results/interpretation, which make it really hard to follow.*

Our approach has been to separate technical implementation (Sec. 2) and scientific background (Sec. 3). We regret that we, indeed, missed some wrong positioning and repetition of statements. We tried to improve the structure of Sec. 3. The detailed description of [7]Be and RH is key information which prohibits further shortening.

*Line 284: "The specific activity per intrusion has changed over the years ..." why?*

This entire part was deleted since it belongs to the [7]Be section.

*Line 347: what is a short gap*

Changed to "short ozone data gaps".

*Lines 361-363: Move this statement somewhere upfront.*

[7]Be was already mentioned in the introduction. Here, we present more details which also requires introductory remarks.

*Lines 387-402: could be summarized by one or two sentences.*

We shortened this part, but not as much as suggested.

*Lines 407-411: isn't it trivial when 7Be is steadily increasing as pointed out in line 378? Remove Fig. 4?*

Figure 4 is necessary to explain the choice of the percentiles and to understand the analysis presented later.

*Line 435: " The monthly minimum RH ..." This is the lowest half-hourly RH reading for each month, right? Is this a useful number? I don't think so. You even state that it might be subject to different sensors. Remove from Fig. 6.*

This is a key statement of the paper. Showing the monthly minim RH is important to identify the capability of the instrumentation of reproducing the extremely dry stratospheric air masses. As discussed later the minimum RH in stratospheric layers is too high in summer which leads to missing STT events when applying the "RH criterion". We believe that the layers are substantially drier than shown by the RH measurements and, thus, the air is significantly more stratospheric than indicated by the humidity data. We see orographic vertical transport as the main reason for the in part excessive RH values. This is an important fact for all mountain stations! The [7]Be criterion should be more reliable.

*Line 485: did Yuan et al. measure in a tunnel? Does that make sense?`*

The IFU measurements were carried out in a window of a tunnel in the rocks above UFS. Here, the lowest influence of upstreaming PBL air was found during daytime. The window is now mentioned.

*Line 718: Stohl, 2001: reference does not exist in the bibliography.*

Not true: the paper is listed in the list of references and the citation is correct.

*Lines 718-720: did you compare with CO data from IAGOS?*

No! This would be an interesting (future) effort yielding information on a global scale.

*Line 750: Claude, 2003: reference does not exist in the bibliography.*

We do not agree: It is listed in the list of references. Unfortunately, the url has changed. We inserted the new one.

*Lines 756 ff: "... increasingly wider layers have separated from the range just above the tropopause ..." Where is this conclusion coming from? Add reference.*

This was done in the preceding sentence. We conclude this ourselves.

*Table 1: What does year "1970.50" etc. mean? Simply write "1970" ...?*

We agree! This offset is due to copying the table from the work sheet of the figure.

*Table 1 and Figs. 14 and 15: "annual ozone averages in intrusions ...". Maybe I missed it in the manuscript but these numbers have to be understood as "annual mean contributions to the ozone levels", correct?*

Yes! Changed!

*Figure 12, caption: criterion needs to read "RH-criterion", not "7Be-RH-criterion", right?*

Changed!

*Fig. 14: this figure (along with the figure caption) is very confusing when first looking at the figure without reading the whole manuscript as it is makes the reader think that O3 is depleted during stratospheric intrusions (since numbers here are lower than in e.g. in Fig. 1 top panel).*

Figure 14 is clearly described in the caption. I do not understand why Reviewer 1 sees an ozone depletion here.

Instead, we think that Fig. 13 is confusing and we removed it and the corresponding text. This figure was prepared for examining the coherence of the seasonal cycle. Too many words are needed to explain the figure.

*Figs. 16, 17, 18 and 20: reword captions to make it clear what is shown (O3 mole fractions at Zugspitze and their individual contributions).*

Changed!

*Fig. 2: it is textbook knowledge that O3 and 7Be rise while rh drops during stratospheric intrusions. Does it really need a case study figure here?*

Not all readers are experts. I think Fig. 2 is an excellent example for illustrating what the Zugspitze data look like and to give an idea about the analysis. Most importantly, a $^{7}$Be rise not necessarily means a decrease in RH (as explained in the text)!

*Figs. 6 and 7: delete one of them or move at least to the sup mat.*

The discussion of RH is essential for judging the quality of the analysis. Figure 6 is needed for demonstrating that low RH values do exist. Without this figure it is not that easy to understand the text. Fig. 7 was prepared to visualize the negative RH trend. We revised Sect. 3.4 for more clearness.

We think that adding a file with supplementary material is bad style which should be limited to journals with page limit. ACP accepts full-size papers. We have never had any problems with publishing papers with up to 20 figures.

*Figs. 8 and 9: delete one of them or move at least to the sup mat.*

Both figures were removed since they can be explained on the basis of Figs. 6 and 7. However, we think that a visualization of the trend change is important. This is now achieved by adding the smoothed curves in Fig. 9 to Fig. 7. The values of the earliest years are not included for clearness: When preparing the combined figure we quite surprisingly found a strong RH discrepancy during that period that would result in curve crossings. This discrepancy now explains the low UFS values during the first years at the bottom of Fig. 18!

*Figs. 16 and 17 are pretty much redundant.*

Figure 16 shows the seasonal cycles, but not all smoothed curves in order to avoid confusion. Fig. 17 shows all relevant annual averages.

**Reply to Review 2**

*This presents an extensive analysis of ozone and related measurements at an important mountaintop site in Europe. The topic certainly appropriate for ACP. Unfortunately I think that the manuscript needs major revision before acceptance to ACP.*

*The principal issue is that in its current form the manuscript is much too long, contains too many figures (many whose value is not clear) and reads like a rambling conference presentation, with too much discussion of some aspects of the analysis while important details are omitted. Since the authors have published many excellent papers in the past, I have no doubt that a rewrite will fix these issues and present a tight, well-reasoned argument for their interesting conclusions.*

Certainly, shortening is possible and, as pointed out above, we have been able to achieve this. Based on the comments we felt motivated to shorten the manuscript and to add clearness, and we thank for the good suggestions. We removed Figs. 8 and 9 (see above) and, most importantly, the complex Fig. 13 However, our intension has been a comprehensive description including background information, in particular on $^7$Be and RH. ACP accepts full-size papers and the corresponding author has successfully published large atmospheric papers with up to 20 figures for more than 20 years.

*Detailed remarks:*

*Line 30: The Introduction is rather long, and contains a lot of (albeit interesting) information that is not needed to set out the scientific question(s) to be addressed by the research.*

Our manuscript describes a rather fundamental effort. Thus, a comprehensive description (review) of the field is mandatory. This manuscript is the final step in a series of studies of our group carried out over more than a dozen years. Some of the information was removed since it belongs to later sections.

*Lines 64-74: The 10 ppb value for the 19th century has been discounted for many years by modelers, who point out that there is no way to have so little ozone in the troposphere unless trees*

*weren't producing isoprene in the 19ᵗʰ century. The Tarasick et al. (2019) paper reviewed many other observations that together suggest a value of 25 ppb for rural (background) air. This paragraph, like many others, seems unnecessary to the point of the paper.*

This is what we wrote. The numbers are really crucial for understanding and for classifying our results. This is obvious from the discussions of Scheel´s results here.

*Lines 75-81: More to the point, more STT is expected, because it is a function of the Brewer-Dobson circulation, which is expected to increase with climate change (Butchart et al., 2006; Butchart, 2014).*

We added a sentence on these papers, thank you!

*Lines 89-94: Osman et al. (2016) attempted to address this question theoretically.*

This is an interesting paper! However, the five papers cited here are preferred because they yield extensive observational material. To repeat the full discussion on mixing would be too much. We decided to cite Osman et al. in the Discussion section.

*Line 214: "standard deviation of the mean" is not a statistical measure. Do you mean standard error of the mean?*

Indeed, this does not make sense (was presumably copied). Changed!

*Lines 219-221: "The IFU and DWD 7Be data agree rather well." The DWD data are mentioned here without introduction. But apparently they were scaled to the IFU data, and "It could not be clarified why this is the case." What is that supposed to mean? It does not inspire confidence in the results.*

See reply to first review: Changed!

*Lines 257-258: "We prefer the "reanalysis" mode that have better explained our observations." This is an odd scientific justification. Do you mean you used reanalysis meteorological fields rather than forecasts? The former would indeed be more accurate.*

We discussed this issue in previous papers. The HYSPLIT "reanalysis" field seems to be less well resolved. Nevertheless, we found in studies on many years of measurements that the agreement with our observations was significantly better. This may have changed as indicated in a new study on stratospheric aerosol to be published within the next few weeks. We added a citation.

*Line 265: This entire section reads like a rambling discussion of the authors' thinking about different thresholds for data filtering, but does not offer any clear justifications for the numbers chosen. In the last paragraph (lines 324-327) the authors seem to be saying that the high 7Be numbers on July 27ᵗʰ must indicate older stratospheric air, but the argument is not clear.*

The justification was given by Trickl et al. (2010). Here, we base our choice on the old results. However, it is a good idea to shorten this section.

*Line 344: Rather than this unscientific justification (the values "look somewhat low"), refer to the discussion of Fabian and Pruchniewicz' data in Tarasick et al. (2019).*

Accepted! A justification is given in Sec. 5.2 anyway.

*Lines 407-411: I really can't make head or tail of what Figure 4 is about. This entire discussion (lines 360-422) seems to be solely for the purpose of justifying using percentile changes in 7Be*

*data, which are clearly a more robust indicator given the other variables involved. This section could be reduced to one or two sentences.*

This figure is needed to understand the algorithm applied to evaluate the "indirect" contribution (Sec. 4). This is now explained.

*Equation 1 is given without derivation or explanation. The last term, a quantity to the $8^{th}$ power, seems absurd. This section (3.4) also seems to go into unnecessary detail about the variability of sensors and the like. RH data are typically of doubtful absolute accuracy, but like the 7Be data, should be useful as an indicator of changes.*

The 8th power was estimated by playing with this correction. We add an explanation.

*Equation 2 looks fairly simple, but the explanation is quite complicated.*

This why we need Fig. 4! The final version of the equation was not that easy to reach. We tried to improve the explanation.

*Lines 631-634: Too many significant figures! Your trends are not accurate to 0.001%.*

The uncertainty of the trend is higher than that of the data because of the correlation between the fit parameters. Thus, it makes sense to add a few decimals to the parameters. It is difficult to judge how many decimals are needed. Nevertheless, we slightly shorten the length of the numbers, now equal in the entire manuscript.

*Figures 11 & 12 show similar trends, but apparently little correlation. That suggests that the trend agreement is coincidental. Please put them on the same horizontal scale so the reader can tell if they do agree. Wait, you did that in Figure 14. Please DELETE Figures 11 & 12!*

There is an important difference between Figs. 11-12 and Fig. 14: In the first case we discuss the numbers of intrusions, in the second one ozone. There is a clear difference in trend: The rise of the ozone mixing ratio is much more pronounced. The agreement of the trends is not just fortuitous. It is obvious that the data filtering identifies the same intrusion cases for both criteria except for limited cases for the RH criterion in which the minimum RH stays above 30 % because of the orographic influence. We added a statement.

*In summary, please reduce the length of the text and the number of figures considerably. There is an interesting message in this analysis but it is difficult to find.*

We tried to follow this recommendation, but there are limits. We added a few sentences on the motivation, in particular in the Discussion.

*References*

*Butchart, N., A.A. Scaife, M. Bourqui, J. de Grandpré, S.H.E. Hare, J. Kettleborough, U. Langematz, E. Manzini, F. Sassi, K. Shibata, D. Shindell, and M. Sigmond, 2006: Simulations of anthropogenic change in the strength of the Brewer-Dobson circulation. Clim. Dyn., 27, 727-741, doi:10.1007/s00382-006-0162-4.*

*Butchart, N. (2014), The Brewer-Dobson circulation, Rev. Geophys., 52, doi:10.1002/2013RG000448.*

*Osman, M.K., W. Hocking and D.W. Tarasick (2016), Parameterization of large-scale turbulent diffusion in the presence of both well-mixed and weakly mixed patchy layers, Journal of*

*Atmospheric and Solar-Terrestrial Physics, 143-144 , 14-36, https://doi.org/10.1016/j.jastp.2016.02.025.*